# HYPOTHESIS SEARCH:
# INDUCTIVE REASONING WITH LANGUAGE MODELS

**Ruocheng Wang**[1*]**, Eric Zelikman**[1*]**, Gabriel Poesia**[1]**,**
**Yewen Pu**[2]**, Nick Haber**[1]**, Noah D. Goodman**[1]
[1] Stanford University, [2] Autodesk Research

## ABSTRACT

Inductive reasoning is a core problem-solving capacity: humans can identify underlying principles from a few examples, which robustly generalize to novel scenarios. Recent work evaluates large language models (LLMs) on inductive reasoning tasks by directly prompting them yielding "in context learning." This works well for straightforward inductive tasks but performs poorly on complex tasks such as the Abstraction and Reasoning Corpus (ARC). In this work, we propose to improve the inductive reasoning ability of LLMs by generating explicit hypotheses at multiple levels of abstraction: we prompt the LLM to propose multiple abstract hypotheses about the problem, in natural language, then implement the natural language hypotheses as concrete Python programs. These programs can be verified by running on observed examples and generalized to novel inputs. To reduce the hypothesis search space, we explore steps to filter the set of hypotheses to implement: we either ask the LLM to summarize them into a smaller set of hypotheses or ask human annotators to select a subset. We verify our pipeline's effectiveness on the ARC visual inductive reasoning benchmark, its variant 1D-ARC, string transformation dataset SyGuS, and list transformation dataset List Functions. On a random 100-problem subset of ARC, our automated pipeline using LLM summaries achieves 30% accuracy, outperforming the direct prompting baseline (accuracy of 17%). With the minimal human input of selecting from LLM-generated candidates, performance is boosted to 33%. Our ablations show that both abstract hypothesis generation and concrete program representations benefit LLMs on inductive reasoning tasks.

## 1 INTRODUCTION

Inductive reasoning – the ability to infer general principles from specific examples and apply them to novel situations – is a core aspect of human intelligence (Peirce, 1868). Recently, large-scale pre-trained language models have received significant interest for their performance across a diverse range of reasoning tasks such as commonsense, arithmetic and symbolic reasoning (Rajani et al., 2019; Shwartz et al., 2020; Nye et al., 2021; Wei et al., 2022; Marasović et al., 2021; Lampinen et al., 2022; Zelikman et al., 2022; Zhou et al., 2022). There has been extensive discussion of language models' impressive "in-context learning" capabilities, a form of inductive reasoning. However, other work suggests that in-context learning of these models has a highly limited capacity to perform inductive reasoning tasks where precise behavior is required (Chollet, 2019; Johnson et al., 2021).

The Abstraction and Reasoning Corpus (ARC) is a particularly challenging inductive reasoning benchmark (Chollet, 2019). For each task in ARC, models are given a set of training input-output pairs with a shared transformation rule, and the goal is to predict the corresponding output(s) given the novel test input(s), as illustrated in Fig 2 (a). ARC is interesting because the answers are fairly natural for humans yet require a complex and precise transformation. Evaluations of LLMs on ARC (Xu et al., 2023b; Mirchandani et al., 2023; Gendron et al., 2023) have directly prompted LLMs to predict outputs by in-context learning, finding poor performance relative to humans (Chollet, 2019; Johnson et al., 2021).

We instead take inspiration from Bayesian models of human inductive reasoning (Tenenbaum et al., 2006; Goodman et al., 2008). That research frames inductive reasoning as posterior prediction: an ideal Bayesian learner assumes a large hypothesis space of possible rules, uses Bayes' rule to form a posterior distribution over hypotheses from examples, then responds accordingly with a posterior-predictive distribution. Studies of human inductive learning have found that people likely

---

*These authors contributed equally to this work

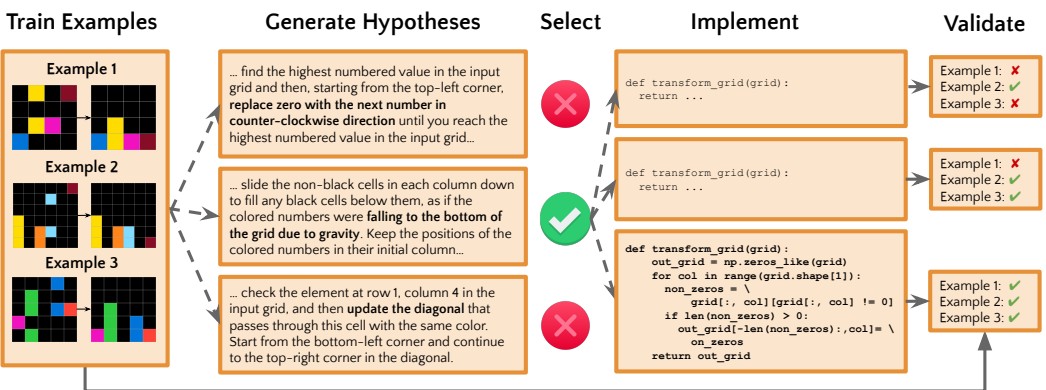

Figure 1: **An overview of our pipeline.** From left to right, starting from a task in the dataset, a language model 1) generates a set of candidate hypotheses, 2) selects a subset, 3) implements each hypothesis in code as a function, and 4) validates the implementations against the training examples.

approximate the full posterior with just a few hypotheses (Vul et al., 2014). Furthermore, people often represent hypotheses of the world at multiple levels of abstraction (Tenenbaum et al., 2011), with more abstract hypotheses guiding the search for more specific ones (Goodman et al., 2011).

We thus propose an approach that improves the inductive reasoning ability of LMs by decomposing the task via hypothesis formation at two levels of abstraction: first by generating hypotheses in natural language and then by realizing these as specific programs that are used for making predictions. Natural language provides abstract representations that uncover key features but are difficult to verify and potentially ambiguous. Programmatic hypotheses are directly verifiable on examples via execution and can naively generalize to new inputs but involve many implementation details that can be distracting to a language model. In other words, we use particular programmatic implementations to act as a precise, generalizable representation of a given inductive hypothesis formulated in natural language. Our pipeline thus disentangles inductive reasoning tasks primarily into two capabilities: the ability to propose accurate natural language hypotheses and the ability to formalize them as programs.

However, in practice LLMs are not yet able to find a good hypothesis with one try. Sampling multiple hypotheses and multiple programs per hypothesis turns out to be sufficient, but can be extremely costly. Thus, we also investigate approaches to reduce the number of hypotheses that must be considered. First, we use an LLM to summarize multiple hypotheses into a smaller number of hypotheses. Second, we experiment with querying a human oracle to go through all hypotheses and indicate which can be ignored. The latter can be viewed as a lower bound on performance that would be achieved by our approach without filtering, because we also find that programs which are correct on all examples almost always generalize correctly, an interesting feature of complex inductive reasoning domains.

We conduct experiments on four inductive reasoning datasets: the Abstraction and Reasoning Corpus (ARC), the one-dimensional variant of ARC (1D-ARC), the Syntax-Guided Synthesis (SyGuS) dataset, and the List Functions dataset. Our results indicate that explicit hypothesis formation substantially improves performance over the direct prompting (ICL) approach. Ablation studies suggest both levels of abstraction – natural-language hypothesis generation and programmatic hypothesis representations – are beneficial to performing inductive reasoning tasks.

**Contributions.** We summarize the contributions of our paper as follows:

- We propose a pipeline that uses language models to solve inductive reasoning tasks by generating and testing hypotheses in natural languages and code.
- We conduct experiments to demonstrate our pipeline achieves significant improvement over baselines on four inductive reasoning tasks across different domains.
- We explore and analyze techniques for reducing the hypothesis search space.

## 2 METHOD

### 2.1 PROBLEM STATEMENT

We consider inductive reasoning tasks that require discovering an underlying transformation rule given input-output examples that follow this unknown rule. More formally, we are given a set of training examples $(x_1, y_1), (x_2, y_2), \ldots, (x_n, y_n)$ where each $y_i = f(x_i)$ for some unknown function $f$. Our goal is for the model to infer the outputs $y'_1, y'_2, \ldots, y'_n$ for a list of novel inputs $x'_1, x'_2, \ldots, x'_n$

---

**Algorithm 1:** Implementing a Python Program from a Natural Language Hypothesis

---

**Input:** Training examples $\{(x_1, y_1), \ldots, (x_n, y_n)\}$, natural language hypothesis $L$, maximum
        number of feedback iterations $N_{\text{feedback}}$, initial LLM prompt template $m$, number of
        programs per hypothesis $K$

**Output:** A Python program $p$ that is expected to be consistent with the training examples and
        hypothesis $L$

$P \leftarrow \text{LLM}(m.\text{format}(L, \{(x_1, y_1), \ldots, (x_n, y_n)\}), n = K)$   `// Generate K programs`
**foreach** program $p \in P$ **do**
    **if** $\forall (x_i, y_i) \in \{(x_1, y_1), \ldots, (x_n, y_n)\} : p(x_i) = y_i$ **then**
        **return** $p$   `// If a program succeeds on all examples, return it.`
**for** $i = 1$ **to** $N_{\text{feedback}}$ **do**
    **foreach** program $p \in P$ **do**
        **for** $(x_i, y_i) \in \{(x_1, y_1), \ldots, (x_n, y_n)\}$ **do**
            $e \leftarrow \text{CatchException}(p(x_i))$
            **if** $e \neq$ null **then**
                $m.\text{append}(p, e)$     `// Add a caught exception to the prompt`
                **break**
            **else if** $p(x_i) \neq y_i$ **then**
                $m.\text{append}(p, x_i, y_i, p(x_i))$     `// Add failed example to prompt`
                **break**
        $p' \leftarrow \text{LLM}(m)$              `// Generate one revised program`
        **if** $\forall (x_i, y_i) \in \{(x_1, y_1), \ldots, (x_n, y_n)\} : p'(x_i) = y_i$ **then**
            **return** $p'$
        $p \leftarrow p'$
**return** $\arg\max_{p \in P} |\{(x_i, y_i) : p(x_i) = y_i \text{ and } \text{CatchException}(p(x_i)) = \text{null}\}|$

---

that captures the transformation $f$. This formulation applies to all four datasets we consider in the experiment settings, as shown in Figure 2. This task is widely studied in program synthesis literature (Acquaviva et al., 2022; Odena et al., 2020; Ellis et al., 2023; Xu et al., 2023a), where a program written in a manually-designed Domain-Specific Language (DSL) is used to represent the transformation, which is applied to the test inputs to obtain the predicted outputs. Recently, there are also multiple works (Webb et al., 2022; Xu et al., 2023b; Mirchandani et al., 2023; Gendron et al., 2023) that do not predict the rule explicitly. Instead, large language models are used to predict the output for novel input examples directly given the training input-output pairs.

## 2.2 OVERVIEW

As illustrated in Figure 1, in our pipeline, we first prompt an LLM to generate hypotheses about the transformation rule shared across the input-output pairs in natural language. We then filter out a smaller set of hypotheses, using either an LLM or human annotator – the goal of this step is simply to reduce the computational cost of later steps. The filtered hypotheses are used to prompt an LLM to generate programs that take in an input example and output the transformed result. These programs are then tested against the initial training examples. Note that, in these domains, we observed that programs that successfully generated outputs for training pairs almost always generalized to test items.

## 2.3 GENERATING HYPOTHESES

The first step in our pipeline is to prompt a language model to generate natural language hypotheses for inductive reasoning problems. For each problem, we provide GPT-4 with a description of the task setup and the problem-specific input-output examples and prompt it to generate hypotheses about possible underlying rules or patterns that could explain the transformation in the given examples. We also provide two problems with human-annotated hypotheses as few-shot demonstrations in the prompt. More precisely, when doing an ARC task, we provide GPT-4 with the input-output examples in the form of a grid of numbers and specify the corresponding colors for each number as part of the prompt, with the exact prompt included in the Appendix A. We sample multiple responses from GPT-4, with a temperature of 1.0, as the hypothesis candidates.

## 2.4 REDUCING NUMBER OF CANDIDATE HYPOTHESES

Ideally, we would like to directly test all of the generated hypotheses by implementing them as Python programs. However, given a potentially large number of hypotheses, testing all of them can be expensive. Thus, we investigate several methods to identify the most promising hypotheses from a set of proposals. For an end-to-end approach, we investigate using LLMs to summarize the full set

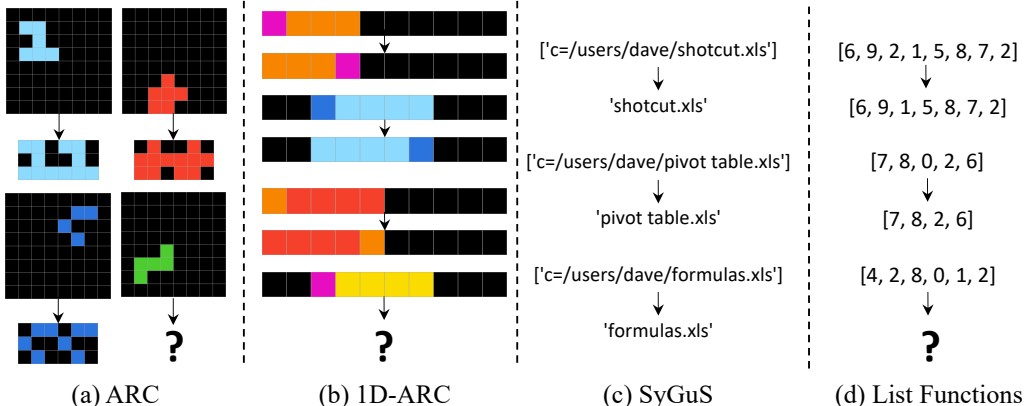

|     |     |     |     |
| --- | --- | --- | --- |
| (a) ARC | (b) 1D-ARC | (c) SyGuS | (d) List Functions |

Figure 2: Example problems in each of our four evaluation datasets.

of hypotheses into a smaller number of hypotheses. Specifically, we directly present GPT-4 with all candidate hypotheses and ask it to produce a smaller number of hypotheses summarizing the given candidate hypotheses. In addition, to help estimate a lower bound on performance if we were to test all hypotheses, we ask a human annotator to go through candidate hypotheses and select correct ones, if any.

## 2.5 IMPLEMENTING PYTHON PROGRAMS FROM HYPOTHESES

The pseudocode for this stage is presented in Algorithm 1. After obtaining a set of candidate hypotheses for each problem, we individually use each hypothesis as the input for GPT-4 and prompt it to generate multiple Python programs that implement the described transformation. Then, we run these programs against the problem's original input-output examples (while still holding out the test examples), determining whether they yield correct outputs for each case. If a code implementation correctly generates the outputs for each of the training examples, it is selected for generating the prediction on the test input example. If no implementation passes all of the training examples, we repeatedly ask GPT-4 to revise the implementations according to the execution results on the training set, including error messages and desired outputs, similar to Chen et al. (2023). This leverages research on code repair spanning multiple decades (Schulte et al., 2010; Pang, 2018; Vasic et al., 2019; Rahman et al., 2021, inter alia). If we cannot achieve a program that passes all the training examples after a preset number of feedback rounds, we select the program that passes the most examples for generating the prediction.

## 3 EXPERIMENTS AND RESULTS

### 3.1 DATASETS

We evaluate our approach on four distinct datasets: the Abstraction and Reasoning Corpus (ARC), the one-dimensional variant of ARC (1D-ARC), BUSTLE's Syntax-Guided Synthesis (SyGuS) dataset and List Functions dataset. These datasets offer diverse and challenging reasoning tasks in the domains of 2D grids, number sequences and strings, enabling us to thoroughly assess the inductive reasoning capabilities of our method. We provide examples of tasks in these datasets in Figure 2.

**ARC.** The Abstraction and Reasoning Corpus (ARC), proposed by Chollet (2019), is a dataset designed to assess models' generalizable reasoning capabilities. It is a dataset of 400 training and 400 evaluation problems. Each problem consists of a set of input-output 2D grids that capture a specific underlying rule or pattern such as geometric transformation and object counting. Each example is a grid with $1 \times 1$ to $30 \times 30$ pixels of any of ten colors – note that the input and output grid need not have the same shape. To effectively analyze this task despite the high cost of GPT-4, in our paper, we randomly select a subset of 100 problems from the 400 training problems as the evaluation dataset.

**1D-ARC.** 1D-ARC is a one-dimensional adaptation of the original ARC dataset proposed in (Xu et al., 2023b). Although simpler than the two-dimensional ARC problems, 1D-ARC offers a more controlled setting to investigate the inductive reasoning abilities of language models as they are trained to handle sequential data. We once again select a random subset for evaluation, this time randomly choosing 6 tasks from each of 1D-ARC's 18 categories for a total of 108 problems.

**SyGuS.** The SyGuS dataset in the BUSTLE paper contains 89 tasks that require representing a mapping between pairs of strings as a program (Odena et al., 2020). This task represents the kinds of problems solved by FlashFill (Gulwani, 2011), a feature in Excel that has been widely cited as an influential real-world example of program synthesis (Le et al., 2017).

**List Functions.** The List Functions dataset proposed in Rule et al. (2020) is a cognitive-science-inspired inductive reasoning benchmark that involves mapping a list of numbers to another list of numbers. The transformation covers basic list operations like duplication, and removal, as well as more complex combinations of recursive, conditional, and numerical reasoning (e.g., sorting, computing difference). The dataset has 250 tasks, each with 8 train and 8 test examples.

## 3.2 ARC

**Settings.** We measure the performance of different methods by computing the accuracy of models' prediction on the test input cases[1]. Although the input-output examples are typically visually presented in 2D pixel grids, we convert them to a text format in the style of NumPy arrays. We include the prompt templates in Appendix A.

### 3.2.1 MAIN RESULTS

We compare the direct prompting baseline to different variants and ablations of our pipeline.

**Direct Prompting.** As done in previous work (Xu et al., 2023b; Mirchandani et al., 2023; Gendron et al., 2023), we provide training examples in a prompt and ask GPT-4 to directly infer novel test inputs' output grids.

**Program Only.** In this ablation, we directly prompt GPT-4 to output Python programs for training examples. We generate 64 programs per task, selecting one passing the most training examples to generate test outputs.

| Method | Accuracy (%) |
|---|---|
| Direct | 17 |
| Program Only | 23 |
| Summarized Hypo. | 30 |
| Human-Selected Hypo. | 33 |
| Human-Written Hypo. | 45 |

Table 1: Results of the baseline and variants of our method on the randomly selected 100 ARC tasks. Our method outperforms baselines with or without human supervision.

**Summarized Hypotheses.** For each problem, we first use GPT-4 to generate 64 candidate hypotheses and then ask GPT-4 to summarize 8 hypotheses from the 64 candidates. We then generate 8 programs for each hypothesis, resulting in 64 candidate programs per problem. This is followed by 3 rounds of execution feedback. Note that during our experiments, we found that GPT-4 only generates correct hypotheses for 49 tasks, according to human annotators. Of the 30 tasks solved with summarized hypotheses, 28 had a correct hypothesis before summarization.

**Human-Selected Hypotheses.** We first prompt GPT-4 to generate 64 hypotheses and then ask a human to manually select a correct one of them for each task, if any exist. Then we generate 8 programs for each hypothesis, followed by up to 3 rounds of execution feedback. As we observe a low false-positive rate, this is approximately a lower-bound to evaluating all hypotheses. Here, we only generate programs for the 49 tasks with selected hypotheses. Of these, 33 (67%) led to correct programs.

**Human-Written Hypotheses.** For this version, we leverage the human language annotations from the LARC dataset (Acquaviva et al., 2022) as golden hypotheses. We then generate 8 programs for each hypothesis, followed by 3 rounds of execution feedback. We treat these human-written hypotheses as oracle solutions in order for us to better understand the extent to which this pipeline is separately bottlenecked by hypothesis generation as opposed to program generation. The main results are shown in Table 1. Using a programmatic representation already boosts the performance over the direct prompting baseline by a large margin, from 17% to 23%. Leveraging the summarized hypotheses is also helpful, improving the performance from 23% to 30%. We obtain the best accuracy 33% when generating programs using human-selected hypotheses. This is on par with the version where we directly leverage the golden human-generated hypotheses. For reference, we also evaluated the method from Icecuber (2020), which is the current DSL-based state-of-the-art approach (Mirchandani et al., 2023). Although it does not use a language model, it provides useful context. We found that it answered 43% of our 100 sampled items correctly, slightly underperforming the human-written results.

### 3.2.2 QUALITATIVE RESULTS

We show an example of generated hypotheses and the corresponding programs generated from the considered methods in Fig. 3. We observe that many of the correct hypotheses generated by GPT-4 are similar to the human-written hypotheses in terms of their specificity, although often less concise. Summarized hypotheses can often become vague and ambiguous, which is potentially the reason for degraded performance. Sometimes the correct hypothesis is omitted from the summarized hypotheses. Note, because we prompt GPT-4 to treat the grids as NumPy (Harris et al., 2020) arrays, we observe that GPT-4 tends to leverage various NumPy functions to perform the desired transformation.

---

[1]The ARC challenge officially uses top-3 accuracy, checking if any of three model outputs are correct, but we consider top-1 accuracy, like most related works (Xu et al., 2023b; Gendron et al., 2023; Mirchandani et al., 2023).

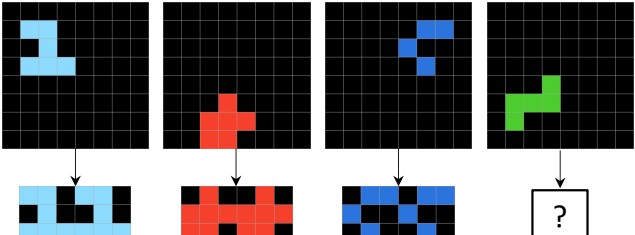



**Summarized Hypothesis and its Corresponding Generated Program**

To make the output, extract the colored shape(s) from the input grid, expand or duplicate them according to the specified pattern, and place the resulting shape(s) into the output grid.

```
1  def transform_grid(input_grid: np.ndarray) -> np.ndarray:
2    unique_numbers = np.unique(input_grid)
3    output_grid = np.zeros((3, 6), dtype=int)
4    for number in unique_numbers:
5      if number == 0:
6        continue
7      num_rows, num_cols = np.where(input_grid == number)
8      min_row, max_row = min(num_rows), max(num_rows)
9      min_col, max_col = min(num_cols), max(num_cols)
10     shape = input_grid[min_row:max_row + 1, min_col:max_col + 1]
11     output_grid[0:shape.shape[0], 0:shape.shape[1]] = shape
12     output_grid[0:shape.shape[0], 3:3 + shape.shape[1]] = shape
13     if shape.shape[0] < 3:
14       output_grid[shape.shape[0]:2 * shape.shape[0], :shape.shape[1]] = shape
15       output_grid[shape.shape[0]:2 * shape.shape[0], 3:3 + shape.shape[1]] = shape
16   return output_grid
```





**Human-Selected Hypothesis and its Corresponding Generated Program**

In the input, you should see a black grid with an 8x8 size. The grid contains one colored shape (either teal, red, or blue), which is a 2x2 square with a corner missing. The output grid size should be 3x6. To make the output, you have to extract the colored shape, and then expand it horizontally by duplicating the entire shape. The result will be a colored shape that is twice as wide as the original one.

```
1  def transform_grid(input_grid: np.ndarray) -> np.ndarray:
2    import numpy as np
3    # Find the non-zero elements and their indices
4    non_zero_indices = np.argwhere(input_grid != 0)
5    min_row, min_col = np.min(non_zero_indices, axis=0)
6    max_row, max_col = np.max(non_zero_indices, axis=0)
7    # Extract the colored shape
8    shape = input_grid[min_row:max_row+1, min_col:max_col+1]
9    # Expand the shape horizontally
10   expanded_shape = np.hstack((shape, shape))
11   return expanded_shape
```



Figure 3: An ARC example of generated hypotheses using different methods and their corresponding generated programs. The summarized and human-selected hypotheses from LLM-generated candidates both yield correct programs. Note, these are `gpt-4-0314`-generated and we omit empty lines.

### 3.2.3 MORE ABLATION STUDIES

**Chain of Thought (Wei et al., 2022).** For this ablation, we wanted to understand the effect of the intermediate language without programs. We found GPT-4's performance dropped to 19%, regardless of whether it generated the intermediate hypothesis or whether human-written hypotheses were used.

**Execution Feedback.** The results of models using different numbers of execution feedback iterations are summarized in Table 2. Execution feedback plays an important role regardless of how hypotheses are generated. However, the performance gain plateaus as the number of feedback iterations increases.

| Method | # feedback iterations | | | |
|---|---|---|---|---|
| | 0 | 1 | 2 | 3 |
| Summarized Hypo. | 24 | 28 | 28 | 30 |
| Human-Selected Hypo. | 26 | 31 | 33 | 33 |
| Human-Written Hypo. | 38 | 44 | 45 | 45 |

Table 2: Accuracy (%) of GPT-4 on ARC using different numbers of feedback iterations.

### 3.3 1D-ARC

In contrast to the ARC experiments, GPT-4's performance on 1D-ARC was notably higher. We observed reasonably correct hypotheses by simply generating 16 hypothesis candidates. Thus, we do not need to obtain a subset of hypotheses to reduce the cost of implementing programs, but instead evaluate on all the hypotheses generated. We compare the direct prompting baseline with our method, with and without natural language hypotheses.

**Direct Prompting** For this experiment, we report the accuracy of direct prompting results from Xu et al. (2023b) on the selected 108 tasks.

**Program Only.** We directly prompt GPT-4 to output Python programs given the training examples. We generate 80 programs for each task and select the program that passed most training examples.

**Full.** We first generate 16 different language hypotheses, then generate 4 programs for each, resulting in 64 programs per problem.

| Method | Accuracy (%) |
|---|---|
| Direct (Xu et al.) | 39.6 |
| Program Only (Ours) | 61.1 |
| Full (Ours) | 73.1 |

Table 3: Experimental results on 1D-ARC. Program and hypothesis generation both contribute to performance improvements.

We summarize our results in Table 3. Generating hypotheses and implementing programs significantly improves the performance on 1D-ARC compared with the direct prompting method.

### 3.4 SYGUS

**Settings.** We evaluate all 89 tasks from the SyGuS dataset. Unlike ARC and 1D-ARC datasets, we follow the convention in the program synthesis literature and treat all examples as training. Accuracy is computed by whether a program passes all training examples.

| Method | Accuracy (%) |
|---|---|
| CrossBeam (Shi et al.) | 74.8 |
| Program Only (Ours) | 94.3 |
| Full (Ours) | 94.3 |

Table 4: Experiment results on SyGuS. Our directly generated programmatic hypotheses and natural-language-conditioned programmatic hypotheses perform similarly.

**Results.** We find that GPT-4 can generate correct programs for 94.3% of the SyGuS tasks using 8 programs with two rounds of feedback without hypothesis generation, demonstrating strong performance in a direct program generation approach. Of the five remaining tasks, we find that three of the tasks have mistakes in their examples. As a result, when using natural language hypotheses to guide the code generation process, GPT-4's performance does not meaningfully change, achieving the same performance 94.3% by generating 4 hypotheses and implementing 2 program for each hypothesis. As a comparison, the state-of-the-art program induction approach CrossBeam (Shi et al., 2022) can solve 74.8% of the dataset using a domain-specific language with 50K program candidates. For reference, the state-of-the-art DSL-based baseline CrossBeam only achieves an accuracy of 74.8 on this dataset.

### 3.5 LIST FUNCTIONS

For this dataset, we sample 100 tasks from it for evaluation. Empirically we found that on this dataset, a correct language hypothesis almost always leads to a correct Python implementation, therefore our Full pipeline will generate 64 hypotheses and implement one Python program for each hypothesis. And the program only ablation will generate 64 programs for each task. The results are summarized in Table 5. Our method consistently outperforms baselines.

| Method | Accuracy (%) |
|---|---|
| Direct | 31 |
| Program Only (Ours) | 59 |
| Full (Ours) | 69 |

Table 5: Experiment results on List Functions. Code and language generation both contribute to performance improvements.

## 4 DISCUSSION

### 4.1 FAILURE CASES

Currently, there are two types of failures in our pipeline. First, the model may be unable to generate a correct and sufficiently precise natural language hypothesis. Second, the model can still generate incorrect programs given a correct hypothesis.

**Hypothesis Generation.** Hypothesis generation is especially challenging for the ARC dataset as it involves recognizing visual patterns in 2D grids. While we observe that GPT-4 has a primitive ability to recognize points, lines, and rectangles, and to identify repetition and symmetry relationships, it has trouble understanding more complex shapes and visual relationships like translation, scaling, and contain-

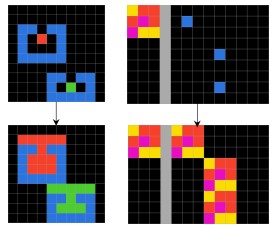

Figure 4: Two examples where GPT-4 has difficulty implementing programs even with correct hypotheses.

ment. This is unsurprising as GPT-4 is trained primarily on text corpora, and the visual grid is input as text in our experiments. Furthermore, we observe that GPT-4 has difficulty proposing reasonable hypotheses for very large grids, possibly due to the limited context length. In contrast, GPT-4 was

quite good at hypothesis generation on 1D-ARC. While the concepts may be easier for this dataset it is certainly the case that the visual encoding is easier. We thus tentatively suggest that current LMs are non-trivially capable of hypothesis generation for inductive learning and anticipate that vision-language models (Driess et al., 2023) may close the remaining gap for visual tasks like ARC.

**Program Generation.** Even with correct hypotheses, difficulties may arise when the task is hard to implement in Python. For example, task 444801d8 shown in Figure 4 was one where the language model failed when given a correct hypothesis. The task is difficult to solve programmatically, even for humans, as it requires identifying an irregular shape and then filling it according to an irregular pattern. This suggests a limitation of using generic Python programs for solving visual inductive reasoning tasks. Natural language hypotheses may also contain ambiguous concepts that mismatch the biases of the program generator. The human-written hypothesis for task 363442ee in Figure 4 is: "In the input, you should see a **color pattern on the left side** and blue squares on the right. The output grid size same size as the input. To make the output, you have to use the blue square as the middle square and recreate the same pattern replacing the blue square with the same color in the middle as the pattern." GPT-4 is unable to understand what "color pattern" refers to and generates an incorrect program by treating the first three columns as the pattern. On the other hand, GPT-4's generated hypothesis mentions that "In the input, you should see a **3x3 colored square** on the left side...", which yields the correct Python implementation. Thus a good match is needed between hypothesis generator and program synthesis, suggesting dircetions for future work.

## 4.2 CONSIDERING EVERY CANDIDATE HYPOTHESIS.

Currently, our pipeline does not consider many candidate hypotheses; we note that this is not a theoretical limitation of our method. In our experiments, we found that when the generated program passes all training cases, it almost always passed the test case (we only observe a single task that is an exception). Therefore, the performance of human-selected hypotheses can reasonably be treated as a lower bound for the performance if we consider every candidate hypothesis. However, we need to sample a large number of (64) hypotheses to have a reasonable hit rate of correct ones, and testing a single candidate hypothesis can take up to 1.5$ (8 programs with two rounds of feedback) – leading us to evaluate summarizing and human filtering. This suggests that the effectiveness of our method will improve automatically as the inference cost of language models decreases.

## 4.3 COMBINATORIAL SEARCH WITH PARSEL

We also explore the application of Parsel, an efficient compositional program generation method (Zelikman et al., 2023), in combination with GPT-4 to enhance the model's ability to generate and evaluate code implementations. This approach aims to capitalize on the benefits of compositional reasoning in problem-solving, by first decomposing a solution, generating multiple implementations of each part of the solution, and then searching over combinations of the implementations. This allows for many more programs to be tested with fewer LLM calls. For human-written hypotheses, this improved performance to 47.5%, but for language-model-generated hypotheses it had the reverse effect. Details can be found in Appendix B.

## 5 RELATED WORKS

**Inductive Reasoning.** Techniques to allow automatic inductive reasoning have been widely studied by the artificial intelligence community as well as the program synthesis community. Given a set of observations, these efforts aim to computationally infer the underlying rules for a set of observations that can be generalized to novel scenarios. Traditional methods usually rely on programs written in manually designed domain-specific languages to represent the rule space and perform searching on the space to obtain the desired program. A number of heuristics have been proposed to speed up the search process. BUSTLE (Odena et al., 2020) proposes a neural search algorithm that takes the intermediate results of partial programs into account during the search process. DreamCoder (Ellis et al., 2023) introduces a wake-sleep algorithm that will dynamically build library functions on top of the primitive operations for solving more complex tasks with less running time. These methods typically require training on a corpora of related tasks, and cannot generalize across different domains due to the limited DSL. Earlier work in this area showed that introducing linguistic knowledge and selecting relevant language descriptions allows for better classifiers (Andreas et al., 2018). Recently, multiple works (Mirchandani et al., 2023; Gendron et al., 2023) tried to evaluate large language models on inductive reasoning tasks. These works directly prompt models to predict the output given the novel input as well as training examples, which leads to poor performance. Our work draws inspiration

from previous program synthesis literature to use programs as representations of the underlying rules. But we instead leverage a general programming language Python, which makes our method applicable to a wide range of different domains such as grid transformation and string transformation.

**Reasoning with Programs.** There has been a consistent effort to introduce program representations into different types of reasoning tasks such as visual reasoning (Andreas et al., 2016; Mao et al., 2019) and question answering (Dong & Lapata, 2016; Zhong et al., 2017). Naturally, these programmatic reasoning works build on prior work on generating programs with language models (Jacob & Tairas, 2010; Schulam et al., 2013; Kushman, 2015; Desai et al., 2016; Chen et al., 2021; Austin et al., 2021; Jain et al., 2022). Programs provide various advantages over end-to-end methods, such as interpretability, generalizability, and efficiency. Mainstream approaches have focused on learning to parse natural language questions into programs of domain-specific languages that can be executed to obtain the answer; a program executor is often jointly learned to execute primitive functions (Andreas et al., 2016; Mao et al., 2019).

Recently, LLMs have been shown to be capable of generating programs written in general-purpose programming languages. This inspired multiple works to leverage LLMs to reason with programmatic representations. Gao et al. (2022) introduced Program-Aided Language models (PAL), and Chen et al. (2022) proposed the "Program of Thoughts" (PoT) prompting, both of which prompt large language models to solve step-by-step math and symbolic reasoning tasks by proposing programs and offload the computation to a Python interpreter. Visprog (Gupta & Kembhavi, 2023) and ViperGPT (Surís et al., 2023) generated programs that can be executed using pretrained perception modules to tackle visual reasoning tasks. These approaches are superior in performance and require minimal data for in-context learning without the need for any training. Notably, the code generated by the models in these papers has primarily served as a computational aid, not a general task representation. In our case, programs serve as testable hypotheses for solving inductive reasoning tasks.

Lastly, Clement et al. (1986) investigated the correlation between analogical reasoning ability and the programming skills of high school students, indicating a significant relationship between the ability to perform analogical reasoning and write compositional programs. Given previously observed parallels between language model behavior and cognitive psychology experiments (e.g., Dasgupta et al. 2022; Aher et al. 2022), language models may exhibit a similar trend.

# 6 LIMITATIONS AND FUTURE WORK

Currently, our method requires multiple LLM queries, which may be costly depending on the complexity of tasks. But, with the rapid improvement and decreasing costs, this allows for increasingly complex tasks to be solved at a given cost. There are also several promising directions for future work building on these results. First, some tasks and objectives are inherently stochastic or challenging to express explicitly as a program – for these, supporting objectives besides exact match is essential (e.g., ROUGE (Lin, 2004)) and may benefit from leveraging from code that leverages other machine learning models (e.g., VisProg (Gupta & Kembhavi, 2023) for visual tasks). However, Python is already notably more expressive than the DSLs that were standard in prior work on program synthesis (Devlin et al., 2017; Ellis et al., 2019; Sharma et al., 2017; Guu et al., 2017).

Lastly, although we observed a low false-positive rate on our datasets (i.e., if a working program was found, it almost always generalized), other inductive reasoning tasks may require further work to avoid false-positives. Under the assumption of a low false-positive rate, which we empirically observed, our method is approximately the same as importance sampling in a hierarchical Bayesian model, where the importance distribution is that of the language model conditioned on the examples. This could potentially guide future work toward more efficient algorithms.

# 7 CONCLUSIONS

In this work, we propose a pipeline that facilitates better inductive reasoning in large language models. The core idea is to first prompt LLMs to generate hypotheses of the underlying rule in natural language, to then implement the hypotheses as Python programs, and to search for programs which can be verified on the training examples and executed on novel inputs for inference. We evaluate the effectiveness of our pipeline on four challenging datasets Abstraction and Reasoning Corpus (ARC), its variant 1D-ARC, a string transformation dataset SyGuS, and List Functions dataset. Our pipeline outperforms the baseline methods by a large margin on all four datasets.

## 8 ACKNOWLEDGEMENTS

This work was supported in part by the Microsoft Accelerate Foundation Models Research program, National Science Foundation Grant No. 2302701, and NSF Expeditions Grant No. 1918771. Gabriel Poesia is supported by a Stanford Interdisciplinary Graduate Fellowship.

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

## A EXPERIMENT DETAILS

**Prompts and Hyperparameters.** For hypothesis generation, The prompts are shown in Figure A.8, Figure A.10 and Figure A.11. We set the temperature to be 1.0 and the maximum number of tokens in response to be 200. For program generation and execution feedback, we use a temperature of 0.7 and set the maximum number of tokens to be 1000. Throughout the experiments, we use `gpt-4-0613` and `gpt-3.5-turbo-0301`. Earlier results used `gpt-4-0314`, which we include in Appendix C.

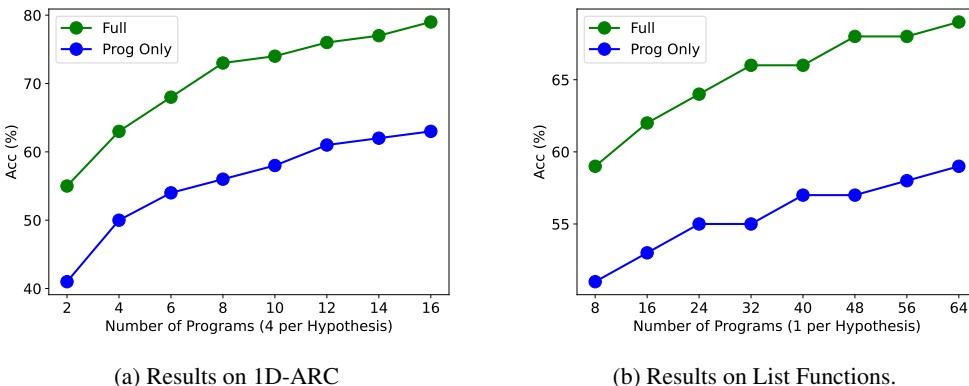

(a) Results on 1D-ARC

(b) Results on List Functions.

Figure A.5: Accuracy of our methods with varying numbers of hypotheses and programs.

**Execution Feedback.** After obtaining programs from language models, we directly execute them on training examples. If there are no programs passing all training examples, we prompt the language model again with the first example that the program fails to pass, and ask it to correct the program. To save cost for experiments on ARC where we generate more than 64 programs, we do not run execution feedback on every program. Instead, we cluster programs by their output on the training examples. Only one program is selected from each cluster for feedback execution.

## B ADDITIONAL RESULTS AND DISCUSSIONS

### B.1 MORE ABLATIONS

**Ablation on the Number of Hypotheses & Programs.**
First, we show the percentage of tasks that have a correct hypothesis when we increase the number of samples in Figure A.6. Increasing the sample size will lead to a steady improvement, although the slope is flattening. We also report how the accuracy changes when we vary the number of hypotheses for each task on ARC-1D and List Func, fixing the number of programs for each hypothesis. For both datasets, our pipeline and the program-only ablation will steadily gain performance improvement as the number of samples increases, as shown in Figure A.5.

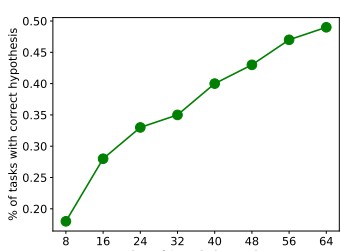

Figure A.6: Percentage of tasks with a correct hypothesis when we increase the number of hypotheses sampled for the ARC experiment.

**Zero-shot Hypothesis Generation.** Here, we explore the effect of examples for hypothesis generation. On 1D-ARC, we remove all the example tasks in the prompt while still generating 16 hypotheses and 4 programs for each hypothesis. This gives an accuracy of 71.3%, which is a little worse than the performance with two-shot prompting 73.1%.

### B.2 GPT-3.5

In this ablation, we leverage GPT-3.5 instead of GPT-4 in our pipeline.

**ARC.** Compared with GPT-4, we find GPT-3.5 mostly generates meaningless hypotheses given the inputs from ARC. We then test GPT-3.5's ability to generate program implementations when given the human-written hypotheses. Because GPT-3.5's context length is only 4096 tokens (GPT-4, which

|  | 1D-ARC | SyGuS | List Func |
|---|---|---|---|
| Direct | 25.9 | N/A | 16 |
| Program Only (Ours) | 23.1 | 80.9 | 46 |
| Full (Ours) | 26.9 | 86.5 | 57 |

Table A.1: Accuracy of GPT-3.5 on 1D-ARC, SyGuS and List Functions datasets.

we use for most of our experiments, has a context length of 8192), only 33 tasks can fit into the prompt. Therefore, we treat the problems that do not fit in the context window as incorrect and do not leverage execution feedback. GPT-3.5 achieves an accuracy of 27% with 128 programs when given human-written hypotheses. Using a 16384 context length version of GPT 3.5 boosts the performance to 30%. This is still worse than GPT-4, but GPT-3.5 is approximately 20 times cheaper than GPT-4. This shows a trade-off between the performance and cost when choosing base models.

**Other datasets.** We continue to evaluate GPT-3.5 on other three datasets, where we found that GPT-3.5 have reasonable performance for both hypothesis generation and program generation. Therefore we keep the same setting and rerun all experiments with `gpt-3.5-0613`. The results are summarized in Table A.1. For 1D-ARC, we generate 16 hypotheses and implement 8 programs for each hypothesis; for SyGuS, we generate 4 hypotheses and implement 2 programs for each hypothesis with two rounds of feedback; for List Functions, we generate 64 hypotheses and 1 program for each hypothesis. All program-only ablation generates the same number of programs in total. We can observe that our method is also effective on GPT-3.5 on a wide range of domains.

### B.3 PARSEL FOR PROGRAM GENERATION

To enhance the performance of program generation, we also adapt a recently proposed method Parsel (Zelikman et al., 2023) to our settings. Instead of directly generating programs, we first generate an intermediate pseudocode program written in Parsel language from a given hypothesis, as shown in Figure A.7. The Parsel language specifies the functions needed to be implemented by specifying the function name, arguments and its desired behavior in natural language. Then the Parsel program is passed to a language model for implementing individual functions.

To allow functions to be implemented with knowledge of their context, unlike the original Parsel paper, we implement all functions needed in a single API call. We then sample multiple trials and extract multiple implementations of each function specified in the Parsel program. Then we will recombine every implementation of each function to generate multiple programs. Using human-written hypotheses, we achieve an accuracy of 47.5% on the 40 randomly selected questions from ARC by generating 4 Parsel Programs for each hypothesis and 8 programs for each Parsel program without any feedback, surpassing the 37.5% accuracy obtained by directly generating programs from hypotheses. However, we found that this yields worse performance with LLM-generated hypotheses: on the 13 selected tasks that GPT-4 can generate correct hypotheses, directly generating 8 programs with 1 round of execution feedback yields an accuracy of 92% while 4 Parsel programs × 8 python programs with 1 round of feedback only yields an accuracy 69%. We suspect that this is due to Parsel introducing a new level of abstraction into our pipeline: given that error might accumulate during the transformation between different levels of abstraction, Parsel increases the probability of generating incorrect final programs. We believe leveraging better code generation techniques is a promising direction to improve our pipeline.

### B.4 PILOT EXPERIMENTS AND NON-SYSTEMATIC FINDINGS

**Specifying the Python Types of Matrices for Grids in ARC.** In the prompt we use for ARC, we indicate the grids are represented as NumPy arrays using Python type hint (`numpy.ndarray[int]`). The typing hint plays an important role in generating programs from language hypotheses, since it encourages LLMs to leverage NumPy functions that are suited for grid transformation, such as flipping, 2D indexing. If we change the typing hint to `List[List[int]]`, LLMs will no longer leverage this library function, which makes the program longer and more error-prone. Using human-written hypotheses, 8 programs and one round of execution feedback. GPT-4 can only achieve 32.5%, compared with the 37.5% performance the using NumPy array signature.

**Using LLMs to Rank Hypotheses.** We also explore to use LLMs to rank language hypotheses to throw away bad hypotheses. This is inspired by Zhang et al. (2023), which reranks code generated from a description based on its probability of generating the description. Because GPT-4 does not expose the log-probabilities of its generated items, and there is no clear way to extract the log-probabilities of the hypotheses, we instead use GPT-3 to rerank the hypotheses generated by GPT-4 by looking at their probabilities of generating the input-output examples, given the hypothesis. We evaluate this ranking method on the 21 tasks where GPT-4 is able to generate a correct language hypothesis from 64 candidates. After the ranking, there are only 10 tasks where the correct hypotheses are placed in the top 16 candidates. This prevents us from reducing the hypotheses needed to implement without sacrificing the overall performance.

**High-Level Representations of ARC Grids.** We observed that many for many tasks in ARC, it is easy for LLMs to come up with reasonable hypotheses if the grids are parsed into a useful geometric representation, such as irregular shapes, diagnoal lines. As a result, we explored the possibility of using alternative geometric representations, similar to concurrent work (Xu et al., 2023b). In particular, we attempted to treat each grid as the result of a sequence of shape placements, for example:

```
Blue Rectangle: (0, 2) size: (4, 5)
Black Line: (2, 4)->(5, 4)
Red Points: [(7, 4), (8, 4), (9, 4)]
```

We implemented an algorithm to identify the shortest possible sequence of shape placements that would result in the observed grid. While we observed that this allowed the model to propose more reasonable hypotheses for a subset of the problems, it harmed performance on more of them. This is due in part to the inherent difficulty of proposing a useful general representation for ARC tasks.

**Potential Data Memorization.** While large language models have shown remarkable performance on numerous benchmarks, there are recurring concerns about whether these models have simply memorized the answers due to observing the problems during training, instead of actually solving the desired tasks. This is particularly true for closed-source models where details of the training set are not publicly available, such as GPT-4. Since the ARC (as well as the LARC dataset with human-written hypotheses) and SyGuS datasets are publicly available on the internet, there is a possibility that GPT-4's training data contains these datasets, which might affect how we interpret these results. While differentiating between memorization and generalization for these close-sourced models remains an open problem, there are few pieces of evidence that show the effectiveness of our method. First, as far as we know, there are no public attempts to solve ARC or SyGuS datasets with Python programs. Second, we tried prompting GPT-4 with some examples in a task and asked it to output other examples in the same task, and GPT-4 failed to do so. Third, the substantial boost of our full pipeline over the direct prediction baseline cannot be simply explained by data memorization.

## C  RESULTS WITH `GPT-4-0314`

We initially ran our experiments with `GPT-4-0314`. Later, due partially to its deprecation, and partially due to a change in our available compute resources, we reran our experiments with `GPT-4-0613`. We include the original result tables here for reference. All the conclusions remain consistent when we scale up the experiments.

| Method | Accuracy (%) |
|---|---|
| Direct | 12.5 |
| Program Only | 17.5 |
| Summarized Hypo. | 27.5 |
| Human-Selected Hypo. | 37.5 |
| Human-Written Hypo.* | 37.5 |

Table A.2: Results of the baseline and variants of our method on the randomly selected 40 tasks from ARC. Our method outperforms baselines with or without human supervision.

| Method | Accuracy (%) |
|---|---|
| Direct (Xu et al.) | 38.8 |
| Program Only (Ours) | 58.3 |
| Full (Ours) | 77.8 |

Table A.3: Experiment results on 1D-ARC. Program and hypothesis generation both contribute to performance improvements.

| Method | Accuracy (%) |
|---|---|
| CrossBeam (Shi et al.) | 74.8 |
| Program Only (Ours) | 94.3 |
| Full (Ours) | 93.2 |

Table A.4: Experiment results on SyGuS. Our directly generated programmatic hypotheses and natural-language-conditioned programmatic hypotheses perform similarly.

---

**Summarized Hypothesis and its Corresponding Generated Parsel Program and Python Program**

To make the output, extract the colored shape(s) from the input grid, expand or duplicate them according to the specified pattern, and place the resulting shape(s) into the output grid.

- - - - - - - - - - - - - - - - - - - - - - - - - - - - - - - - - - - - - - - - - - - - - - -

```
transform_grid(input_grid): Extract figure from input grid and create output grid
--extract_figure(input_grid): Identify the non-zero elements in the input grid and their position
--create_output_grid(figure): Create a 3x6 grid and copy the figure two times into it
```

- - - - - - - - - - - - - - - - - - - - - - - - - - - - - - - - - - - - - - - - - - - - - - -

```
 1 def extract_figure(input_grid):
 2     non_zero_positions = np.argwhere(input_grid != 0)
 3     return input_grid[non_zero_positions.min(axis=0)[0]:non_zero_positions.max(axis=0)[0] + 1,
 4                       non_zero_positions.min(axis=0)[1]:non_zero_positions.max(axis=0)[1] + 1]
 5
 6 def create_output_grid(figure):
 7     output_grid = np.zeros((3, 6), dtype=int)
 8     output_grid[:,:figure.shape[1]] = figure
 9     output_grid[:,3:3+figure.shape[1]] = figure
10     return output_grid
11 def transform_grid(input_grid):
12     extracted_figure = extract_figure(input_grid)
13     output_grid = create_output_grid(extracted_figure)
14     return output_grid
```

Figure A.7: An ARC example of generated hypotheses using Parsel (Zelikman et al., 2023) using the same task as Figure 3.

**Prompt for Hypothesis Generation**

[Role: system]
You will be given a list of input-output pairs. Each input and output is a grid of numbers representing representing a visual grid. There is a SINGLE pattern that transforms each input grid to the corresponding output grid.
The pattern may involve counting or sorting objects (e.g. sorting by size), comparing numbers (e.g. which shape or symbol appears the most? Which is the largest object? Which objects are the same size?), or repeating a pattern for a fixed number of time.
There are other concepts that may be relevant.
- Lines, rectangular shapes
- Symmetries rotations, translations.
- Shape upscaling or downscaling, elastic distortions.
- Containing / being contained / being inside or outside of a perimeter.
- Drawing lines, connecting points, orthogonal projections.
- Copying, repeating objects.
You should treat black cells as empty cells (backgrounds).
The number in the input grid can be mapped to the following colors:0:black; 1:blue; 2:red; 3:green; 4:yellow; 5:grey; 6:fuschia; 7:orange; 8:teal; 9:brown
Output the language description of the transformation.

- - - - - - - - - - - - - - - - - - - - - - - - - - - - - - - - - - - - - - - - - - - - - - - - - -

[Role: user]
Case 0:
Input:
[[3 3 8]
[3 7 0]
[5 0 0]]
Output:
[[0 0 5]
[0 7 3]
[8 3 3]]
Case 1:
Input:
[[5 5 2]
[1 0 0]
[0 0 0]]
Output:
[[0 0 0]
[0 0 1]
[2 5 5]]

- - - - - - - - - - - - - - - - - - - - - - - - - - - - - - - - - - - - - - - - - - - - - - - - - -

[Role: assistant]
Describing the input grid: In the input, you should see a black grid with a colored shape
Describing the size of the output grid: The output grid size is the same as the input grid
Describing how to transform the grid: To make the output, you have to rotate the whole grid two times. Imagine that the entire grid has been flipped vertically and horizontally.

- - - - - - - - - - - - - - - - - - - - - - - - - - - - - - - - - - - - - - - - - - - - - - - - - -

[Role: user]
Case 0:
Input:
[[0 1 0 1]
[0 0 0 1]
[1 0 1 0]
[0 0 0 1]
[4 4 4 4]
[0 2 0 2]
[0 0 0 2]
[2 0 0 2]
[2 2 2 0]]
Output:
[[0 0 0 0]
[0 0 0 0]
[0 0 3 3]
[3 3 3 3]]
...

Figure A.8: The prompt used to generate the hypotheses. Continued in Figure A.9. We use the Chat completions API, where every prompt consists of a list of messages representing a conversation. Each message contains a text and a role indicating the agent producing the text.

```
...
Case 1:
Input:
[[1 1 0 0]
[1 0 1 0]
[1 1 0 1]
[0 1 1 0]
[4 4 4 4]
[0 2 2 2]
[2 0 2 0]
[2 2 2 2]
[2 2 2 2]]
Output:
[[3 0 3 3]
[0 0 0 0]
[0 0 3 0]
[3 0 0 3]]
Case 2:
Input:
[[0 1 0 0]
[1 0 1 1]
[1 1 1 0]
[1 1 1 0]
[4 4 4 4]
[0 0 0 0]
[0 2 0 2]
[2 2 0 2]
[0 2 0 0]]
Output:
[[0 3 0 0]
[3 3 3 0]
[0 0 3 3]
[3 0 3 0]]
Case 3:
Input:
[[1 0 1 1]
[0 0 0 1]
[1 1 0 0]
[0 0 1 1]
[4 4 4 4]
[0 2 2 2]
[0 2 2 2]
[2 0 2 2]
[2 2 2 2]]
Output:
[[3 3 0 0]
[0 3 3 0]
[0 3 3 3]
[3 3 0 0]]
```
- - - - - - - - - - - - - - - - - - - - - - - - - - - - - - - - - - - - - - - - - - - -
[Role: assistant]
Describing the input grid: In the input, you should see a 4x4 square of blue and black boxes on top of a yellow line. Below the yellow line is a 4x4 square of red and black boxes.
Describing the size of the output grid: The output grid size .should be 4x4.
Describing how to transform the grid: To make the output, you have to compare the squares in the top and bottom 4x4 square. If they match (both black or both color) make the output square black. If they do not match, make the output square green.
- - - - - - - - - - - - - - - - - - - - - - - - - - - - - - - - - - - - - - - - - - - -
[Role: user]
Case 0:

...
<Training examples of the task to be solved>
...

Figure A.9: The prompt used to generate the hypotheses (First half in Figure A.8).

Prompt for Program Implementation

[Role: User]
Example 0:
Input:
[[0 0 0 0 0 0 0 0]
[0 8 8 0 0 0 0 0]
[0 0 8 0 0 0 0 0]
[0 8 8 8 0 0 0 0]
[0 0 0 0 0 0 0 0]
[0 0 0 0 0 0 0 0]
[0 0 0 0 0 0 0 0]
[0 0 0 0 0 0 0 0]]
Output:
[[8 8 0 8 8 0]
[0 8 0 0 8 0]
[8 8 8 8 8 8]]
Example 1:
Input:
[[0 0 0 0 0 0 0 0]
[0 0 0 0 0 0 0 0]
[0 0 0 0 0 0 0 0]
[0 0 0 0 0 0 0 0]
[0 0 0 0 0 0 0 0]
[0 0 0 2 0 0 0 0]
[0 0 2 2 2 0 0 0]
[0 0 2 2 0 0 0 0]]
Output:
[[0 2 0 0 2 0]
[2 2 2 2 2 2]
[2 2 0 2 2 0]]
Example 2:
Input:
[[0 0 0 0 0 0 0 0]
[0 0 0 0 0 1 1 0]
[0 0 0 0 1 0 0 0]
[0 0 0 0 0 1 0 0]
[0 0 0 0 0 0 0 0]
[0 0 0 0 0 0 0 0]
[0 0 0 0 0 0 0 0]
[0 0 0 0 0 0 0 0]]
Output:
[[0 1 1 0 1 1]
[1 0 0 1 0 0]
[0 1 0 0 1 0]]
Now, please write a python program transform_grid(input_grid: np.ndarray[int]) -> np.ndarray[int] that transforms the input grid to the corresponding output grid.
Hint: You may want to use the following guidance to implement the function:
Hint: You may want to use the following guidance to implement the function:
**To make the output, extract the colored shape(s) from the input grid, expand or duplicate them according to the specified pattern, and place the resulting shape(s) into the output grid.**
The number in the input grid can be mapped to the following colors:0:black; 1:blue; 2:red; 3:green; 4:yellow; 5:grey; 6:fuschia; 7:orange; 8:teal; 9:brown
Just reply with the implementation of transform_grid(input_grid: np.ndarray[int]) in Python and nothing else, each cell in the output should only be numbers from 0 to 9.

Figure A.10: The prompt used to generate the program given the hypothesis (bold in the text) for the same task as Figure 3.

> **Prompt for Hypothesis Summarization**
>
> [System]
> You are a genius solving language puzzles.
> [User]
> Given a list of rules, categorize them into eight distinct categories based on their similarities. For each category, synthesize the rules into a single, specific rule that combines the ideas of all rules in that category, while clearly differentiating it from the other categories.
> The new rule should be as specific as possible, following the format of the given rules.
> The new rule should be applicable without any information from the original rules - i.e. it should be standalone.
> Rules:
> `{Hypothesis 1}`
> `{Hypothesis 2}`
> ...

Figure A.11: The prompt used to summarize 8 hypotheses from 64 generated hypotheses.

