# OpenReview forum: "Hypothesis Search: Inductive Reasoning with Language Models"
_ICLR.cc/2024/Conference — ICLR 2024 poster_

### Official Review · Reviewer_e7wF · 2023-11-01

**Soundness:** 3 good
**Presentation:** 3 good
**Contribution:** 3 good
**Rating:** 6
**Confidence:** 3

**Summary:**

The authors propose to use large language models (LLM) to generate hypothesis for abstraction and reasoning corpus (ARC).
Given a task in ARC, the LLM first propose a set of hypothesis, then either a language model or a human in the loop can select a subset hypothesis for generating a program that satisfy the hypothesis as the specification.
The automated pipeline which uses the LLM to perform the selections has 27.5% accuracy, and with human in the loop has 37.5%.

**Strengths:**

Originality: 5/5
The idea of using the LLM to generate hypotheses and then synthesizing the downstream Python program is novel and interesting. The experimental result gives positive feedback that the natural language is capable of representing human intuition in this low data in-context learning environment.

Quality: 3/5
The experimental result shows promising improvement in the methodology. However, it seems still quite expensive and not reliable enough to generate 64 different hypotheses for the language model by setting the temperature to 1.0. It would be nice to have a chart on the GPT-4 query number against the rate where it hit the correct hypothesis.

Clarity: 3/5
There are quite a lot of details that are necessary to help understand the work in the supplementary material, for example, the GPT-4 prompts.

Significance: 4/5
This work is important to the program synthesizing community in how to synthesize a natural and intuitive program, instead of synthesizing a functionally correct but not necessarily generalizable program.

**Weaknesses:**

See strength.

**Questions:**

It would be nice if there a statistical analysis on the failure case analysis.

---

> ### Author Response · Authors · 2023-11-21
> **Response to Reviewer e7wF**
>
> Thank you for your encouraging comments and insightful questions!
>
> > It would be nice to have a chart on the GPT-4 query number against the rate where it hit the correct hypothesis.
>
> Great idea; we’ve added this to Appendix B.1. The number of tasks with correct hypotheses increases consistently as we sample more hypotheses from GPT-4.
>
> > It would be nice if there a statistical analysis on the failure case analysis.
>
> Thanks for the excellent suggestion! For the human-selected evaluation, we’ve now added a simple failure case analysis - namely, we were curious about the following: for what subset of the questions did the model fail because it failed to generate a correct hypothesis, and for what subset did it fail because it failed to implement a correct hypothesis? We looked at the 100 sampled tasks and the human-selected correct generated hypotheses (though, we reiterate here that “correct” is both subjective and somewhat arbitrary without experiments in line with those conducted in LARC). For 49% of them, at least one correct hypothesis was identified as correct. Of those selected, 33 (67%) led to correct programs. However, when given all correct language (i.e., the Human-Written Hypothesis evaluation), the model successfully implements 45% correctly. Of these, only one task is solved by the human-selected pipeline that isn’t solved by the human-written pipeline. This may suggest that there is a relationship between the tasks that the model can propose a hypothesis for and the tasks for which it can implement that hypothesis.
>
>
> > However, it seems still quite expensive and not reliable enough to generate 64 different hypotheses for the language model by setting the temperature to 1.0.
>
> We strongly agree with this in principle, and have now highlighted this in our limitations section! However, we note that over time the best large language models do get better, cheaper, and faster, which will make our method more effective. Less than a month ago, a version of GPT-4 was announced that appears to be broadly improved, faster, is over two times less expensive, and (ostensibly) supports a 128k context window.

---

### Official Review · Reviewer_7fhN · 2023-11-02

**Soundness:** 2 fair
**Presentation:** 3 good
**Contribution:** 2 fair
**Rating:** 3
**Confidence:** 4

**Summary:**

This paper proposes a new program synthesis framework for solving inductive reasoning problems based on large language models and prompting techniques. The idea is to first generate hypotheses based on the training samples, and then select a few hypotheses to realize their implementations. The implementations are verified on the training samples and the best implementation is selected to perform inference on the test samples. Experiments on ARC and 1D-ARC verify the effectiveness of the proposed method, while the proposed method doesn't outperform direct prompting on SyGuS.

**Strengths:**

This paper shows that large language models can generate natural language hypotheses based on the training samples. The generated hypothesis can improve the performance of program synthesis on inductive reasoning benchmarks.
The paper conducts experiments on ARC, which is a challenging benchmark for inductive reasoning.

**Weaknesses:**

The overall prompting framework in this paper is very similar to self-debug[1] , except that self-debug focuses on iterative refinement, while this paper emphasizes hypothesis search. If this is the point, the authors should provide a deeper analysis of the generated hypotheses. Algorithm 1 has a similar high-level idea of Figure 3 from the self-debug paper. So this paper is more like revisiting self-debug from a different perspective, which limits its novelty and contribution. This paper also misses an important citation[2].
Experiments results are not sufficient to justify the significance of the method. Of the 3 datasets used in the paper, the proposed method only works on ARC and 1D-ARC, which are very similar. Besides, it only uses 40 samples for inference and the variance of the performance is not reported. It is likely the observation in this paper may be overestimated due to variance in performance and model selection.
The contribution of this paper is not very clear. From the intro, it looks like the authors try to solve the inductive reasoning problem. From the experiments, there is no comparison with non-LLM baselines, and it looks more like an ablation study of using natural language hypotheses in program synthesis.

[1] Chen, et al. Teaching large language models to self-debug. arXiv 2023.
[2] Austin and Odena, et al. Program synthesis with large language models. arXiv 2021.

**Questions:**

Questions:
Is there any deeper connection between Hypothesis Search and the Bayesian learner mentioned in the introduction?
Sec. 2.4. “a lower bound” -> It is not very clear to me why it is a lower bound before I read the experiment section. May rewrite the last sentence.
Sec. 3.1. “It contains” -> incomplete sentence.
Sec. 3.2.1. perf -> per
Sec. 3.2.1. Human-Selected Hypotheses. Why do you use 3 rounds of execution feedback here? The other experiments are based on 2 rounds.
Sec. 3.2.3. How about the ability of GPT3.5 in generating hypotheses? Why is there no table for this section?
Sec. 3.4. Why is there no table for this section? Also the last sentence is an overclaim. It’s the improvement of GPT-4 over CrossBeam, not the proposed prompting technique.

---

> ### Author Response · Authors · 2023-11-21
> **Author Response to Reviewer 7fhN**
>
> Thank you for your constructive comments. We appreciate the chance to clear up any points of confusion, but please do not hesitate to let us know if any further clarifications are needed.
>
> > while the proposed method doesn't outperform direct prompting on SyGuS.
>
> We want to clarify that there is no direct prompting baseline in the SyGuS experiment. because there are no testing examples in SyGuS. Our method performs slightly worse than our program-only hypothesis. We note that on a new evaluation using gpt-4-0613, our full pipeline obtains the same performance with the program only hypothesis.
>
> > The overall prompting framework in this paper is very similar to self-debug[1] , except that self-debug focuses on iterative refinement, while this paper emphasizes hypothesis search.
>
> First, we apologize for the misunderstanding. Self-debug does not explore inductive reasoning (that is, taking a collection of examples and identifying their underlying rule). Not only does it not “emphasize hypothesis search” - indeed, hypothesis search is not part of their method, and is our primary contribution.
>
> In particular, we are (to our knowledge) the first to demonstrate that language models can effectively generate a collection of hypotheses to solve an inductive reasoning problem, and then select the right hypothesis by evaluating its ability to explain the inductive reasoning problem’s examples. We have added a discussion to the Introduction section to help clarify this.
>
> While our paper does use the language model to revise its solutions in code, we also strongly agree that this idea is not one of our contributions. We have updated the paper to more clearly emphasize that our primary contribution is programmatic inductive reasoning pipeline via hypothesis generation, and have added additional citations to prior work on automated code repair [1,2,3,4] to help contextualize this work.
>
> [1] "A Bidirectional LSTM Language Model for Code Evaluation and Repair" Rahman et al 2021
> [2] "Neural Program Repair by Jointly Learning to Localize and Repair" Vasic et al 2019
> [3] "Deep Learning for Code Repair" Pang 2018
> [4] "Automated program repair through the evolution of assembly code" Schulte 2010
>
> > From the intro, it looks like the authors try to solve the inductive reasoning problem. From the experiments, there is no comparison with non -LLM baselines, and it looks more like an ablation study of using natural language hypotheses in program synthesis.
>
> Thank you for highlighting this aspect of our work. First we want to note that on SyGuS, we have reported the accuracy of a non-LLM baseline CrossBeam [5], which is outperformed by our methods. In our paper, we specifically concentrate on exploring inductive reasoning through the lens of language models, as indicated in our title. This focus is intentionally chosen to better understand the capabilities of LLMs in this context. Our main focus was on understanding the impact of generating various kinds of intermediate hypotheses when solving inductive reasoning tasks with language models, a type of reasoning that they have been known to struggle with. For some context, we’ve also now highlighted the special-purpose DSL-based state-of-the-art on ARC (according to [6]) in our text, which performs marginally worse than our results with human-written hypotheses, solving 43 of our 100 sampled tasks.
>
> >  Experiments results are not sufficient to justify the significance of the method. Of the 3 datasets used in the paper, the proposed method only works on ARC and 1D-ARC, which are very similar.
>
> Could you please clarify this? Our results indicate a significant improvement over prior work on all of the datasets. Note that both the program-only and the full hypothesis search methods show significant improvement over prior work on SyGuS. We have now added the cognitive-science-inspired inductive reasoning dataset, List Functions, from “The Child as Hacker” [7] which is fairly different from any of the current three datasets. As on the ARC datasets, this shows a clear advantage with the natural language hypotheses.
>
> | Method           | 64 hypotheses   |
> |------------------|-----------------|
> | Direct Prompting | 31%             |
> | Program Only     | 59%             |
> | Full             | 69%             |

---

> ### Author Response · Authors · 2023-11-21
> **Author Response to Reviewer 7fhN (Cont')**
>
> Other points:
> > This paper also misses an important citation.
>
>  Thank you, we’ve added [8] to the related works. Note that this work focuses on solving program synthesis given natural language descriptions, while our work focuses on inductive reasoning tasks where we only observe a few input-output examples as the specification for programs.
>
> > Besides, it only uses 40 samples for inference and the variance of the performance is not reported.
>
> Thanks for this great point. Originally, scaling this up was prohibitively expensive. However, we have now been able to access additional resources, allowing us to extend these results to 100 tasks for ARC, 1D-ARC and the new List Functions datasets. All the conclusions remain consistent with the previous results, which demonstrate the effectiveness of our proposed pipeline.
>
> > The contribution of this paper is not very clear.
>
> Thank you, we’ve added a paragraph to the introduction to help clarify.
>
> > Is there any deeper connection between Hypothesis Search and the Bayesian learner mentioned in the introduction?
>
> Under the assumption of a low false-positive rate, which we empirically observed, our method is approximately the same as importance sampling in a hierarchical Bayesian model, where the importance distribution is that of the language model conditioned on the examples. This could potentially guide future work toward more efficient algorithms -- we’ve added a note about this to future work.
>
>
>
>
> > Why do you use 3 rounds of execution feedback here?
>
> We have updated experiments to uniformly use 3 rounds of feedback.
>
> > How about the ability of GPT3.5 in generating hypotheses?
>
> For ARC we noted that we observed GPT-3.5 to mostly generate meaningless hypotheses. However, motivated by this question, we ran additional experiments to understand the performance of GPT3.5-generated hypotheses on 1D-ARC, SyGuS and List Functions when using GPT3.5-generated hypotheses. Notably, on all of the datasets we evaluated besides ARC, natural language hypothesis search improved performance over both a direct prompting baseline, as well as a program-only hypothesis search. Surprisingly, in the case of 1D-ARC, the program-only GPT-3.5 results were actually slightly worse than its direct prompting results, while the full hypothesis search pipeline was better than both. We have added the following table to Appendix B.
>
> |                    | 1D-ARC | SyGuS | List Func |
> |--------------------|--------|-------|-----------|
> | Direct             | 25.9   | N/A   | 16        |
> | Program Only (Ours)| 23.1   | 80.9  | 46        |
> | Full (Ours)        | 26.9   | 86.5  | 57        |
>
>
>
> >Sec. 3.4. Why is there no table for this section?
>
> We have updated the paper to summarize the results in a table.
>
> > Also the last sentence is an overclaim. It’s the improvement of GPT-4 over CrossBeam, not the proposed prompting technique.
>
> We agree that the results are not directly comparable. We have revised the text accordingly. We provide the number for CrossBeam as for a reference of the state-of-the-art DSL-based method.
>
> [5] "Crossbeam: Learning to search in bottom-up program synthesis," Shi et al. 2022
> [6] "Large Language Models as General Pattern Machines," Mirchandani, et al. 2022
> [7] "The Child as Hacker," Rule et al. 2020
> [8] "Program synthesis with large language models." Austin, et al. 2021

---

### Official Review · Reviewer_Bu26 · 2023-11-05

**Soundness:** 3 good
**Presentation:** 3 good
**Contribution:** 2 fair
**Rating:** 6
**Confidence:** 3

**Summary:**

This paper prompts LLMs to generate Python programs to solve symbolic pattern recognition problems. This may be better than letting the model directly predict answers. On Abstraction and Reasoning Corpus (ARC) where the inputs are 2D or 1D pixel grids, letting GPT-4 generate natural language hypotheses to then guide program generation improves the result. Hypothesis generation slightly harms the performance on SyGuS where the inputs are strings.

**Strengths:**

1. The paper proposed to let GPT-4 generate programs to solve symbolic pattern recognition tasks. It shows that using natural language hypotheses to guide the program generation can be helpful on ARC, on which letting the model directly generate programs results in bad programs.
1. The paper reports the limitation that on SyGuS where the model can directly generate programs, hypothesis guidance is not helpful.
1. The presentation is clear and several findings are interesting.

**Weaknesses:**

1. The technical novelty is limited and the main challenge of generating high-quality programs is largely unsolved.
2. The effectiveness of the proposed method of using hypotheses to guide program generation has unclear applicability. (1) GPT-3.5 fails to generate meaningful hypotheses. (2) GPT-4 hypotheses are not helpful on SyGuS where GPT-4 can directly generate good programs. (3) GPT-4 hypotheses are helpful on ARC, but ARC results are still only 37.5 with the hypotheses. Practitioners will have to develop alternative models that can better understand 2D geometry to solve the task and then natural language hypotheses may no longer be helpful as in SyGuS. (4) Model-generated hypotheses hurt the performance of Parsel, a compositional program generation method that can significantly  improve the performance when model-generated hypotheses are not used.
3. Multiple questions need to be clarified; some requires experimental results. Please refer to Questions.
4. Typo: Sec 3.1 "It contains Although simpler..."

**Questions:**

1. Sec 3.2.2 says summarized hypotheses can often become vague and ambiguous. Will the hypotheses used to guide program generation be of higher quality if you let the model rank the hypotheses? You could analyze the recall@k, i.e., whether top k hypotheses contain a correct one.
1. ARC: In Table 2, using human written hypotheses only has 37.5 accuracy. Does that mean LLM fails to write programs based on correct hypotheses? The statement at the end of page 5 that "GPT-4 is pretty good at both generating hypotheses and realizing them as programs" requires some more evidence or explanation.
1. ARC: In Table 2, the accuracy with human-selected and human-written hypotheses are both 37.5. Does this mean model-generated hypotheses for each task almost always contain a correct one? Or is it the case that model-generated hypotheses sometimes have mistakes but, when correct, leads to better programs, and thus both 37.5? Can you evaluate the recall of model-generated hypotheses, either by some automatic metric or human evaluation?
1. For ARC, why do you only consider top-1 accuracy but not top-3 as in the official evaluation? Can you compare your method with state-of-the-art methods on the task?
1. What are the types of tasks that (1) program generation and (2) hypotheses search can be helpful? Can you summarize the features of such tasks? "Inductive reasoning tasks" is too general and abstract. To begin with, is it true that the method is applicable only to symbolic pattern recognition tasks?

---

> ### Author Response · Authors · 2023-11-21
> **Author Response to Reviewer Bu26**
>
> Thank you for the thorough and very helpful questions!
>
> > Hypothesis generation slightly harms the performance on SyGuS where the inputs are strings…
>
> Thank you for this point! We’d like to clarify that we don’t consider the SyGuS results as negative: indeed, our pipelines (both Program Only and Full) both perform significantly better than prior work, and we’ve added a table to Sec. 3.4 to help highlight this. However, we do find that the natural language hypotheses are not a useful additional abstraction in this dataset. This is likely because the SyGuS tasks are relatively simple so the Program Only ablation already achieves saturated performance.
>
> However, we also introduce a new dataset to demonstrate the usefulness of natural language hypotheses on non-ARC tasks. Specifically, we now investigate the cognitive-science-inspired inductive reasoning dataset, List Functions, from “The Child as Hacker” [1] which is fairly different from any of the current three datasets. As on the ARC datasets, this shows a clear advantage with the natural language hypotheses.
>
> | Method           | 64 hypotheses   |
> |------------------|-----------------|
> | Direct Prompting | 31%             |
> | Program Only     | 59%             |
> | Full             | 69%             |
>
> Due to a change in our available resources, we also updated our results with a slightly newer version of gpt-4, gpt-4-0613. With the updated model, both pipelines correctly solve 94.3% of the examples.
>
> [1] “The Child as a Hacker,” Rule et al. 2020
>
> > The main challenge of generating high-quality programs is largely unsolved.
>
> We agree that there is still substantial work to be done before these difficult inductive learning tasks can be considered complete. On the other hand, our methods achieve large improvement over four datasets, which we believe is substantial. Our main contribution is showing how to approach inductive learning using program generation as a component, instead of tacking the program generation problem. Future improvements on generating high-quality programs can be expected to benefit our approach to induction reasoning tasks.
>
> >The technical novelty is limited
>
> Novelty is, of course, a subjective quality. Some technical novelty derives from the extensive mathematical reasoning required. Other novelty derives from simple ideas which haven’t been previously noticed and yet have powerful implications. We believe our paper is an example of the latter. The ability to do inductive learning from a few examples has been a cornerstone ability of LLMs in recent years, and their failure in more complex inductive learning tasks (e.g. ARC) has been taken by a substantial subset of computer scientists as a condemnation of the enterprise. Thus finding that a structured approach to induction yields very large improvements, and that this depends on the right staging between natural and formal languages, is an important finding. To those who expected LLMs to make no progress on hard problems of induction it is a surprising, we’d say novel, one.

---

> > ### Author Response · Authors · 2023-11-21
> > **Author Response to Reviewer Bu26 (Cont')**
> >
> > > The effectiveness of the proposed method of using hypotheses to guide program generation has unclear applicability. (1) GPT-3.5 fails to generate meaningful hypotheses. (2) GPT-4 hypotheses are not helpful on SyGuS where GPT-4 can directly generate good programs. (3) GPT-4 hypotheses are helpful on ARC, but ARC results are still only 37.5 with the hypotheses. Practitioners will have to develop alternative models that can better understand 2D geometry to solve the task and then natural language hypotheses may no longer be helpful as in SyGuS. (4) Model-generated hypotheses hurt the performance of Parsel, a compositional program generation method that can significantly improve the performance when model-generated hypotheses are not used.
> >
> > Thanks for these questions! We’ll respond to them one-by-one:
> > 1. While GPT-3.5 underperforms GPT-4 (which should not be surprising), given the far lower cost, as well as the recent release of another, better turbo model, this difference should continue to decrease with time. However, motivated by this question, we ran additional experiments to understand the performance of GPT3.5-generated hypotheses on 1D-ARC, SyGuS and List Functions when using GPT3.5-generated hypotheses. Notably, on all of the datasets we evaluated besides ARC, natural language hypothesis search improved performance over both a direct prompting baseline, as well as a program-only hypothesis search. Surprisingly, in the case of 1D-ARC, the program-only GPT-3.5 results were actually slightly worse than its direct prompting results, while the full hypothesis search pipeline was better than both. We have added the following table to Appendix B.
> > 2. Please refer to our response to your first question above.
> > 3. We agree that our method cannot achieve perfect accuracy on ARC, a dataset notably well-known for its difficulty. We believe our proposed pipeline is a first step towards using language models for solving related reasoning tasks, which beats the baseline method by a large margin on ARC. Future work can explore the directions of generating better hypotheses or generating better programs from language hypotheses.
> > 4. We agree Parsel does not perform optimally with our existing pipeline. It is important to note that Parsel was initially developed specifically for solving competitive programming problems from human written descriptions. The mismatch seems to indicate differences in the decomposability of human and model-generated hypotheses, which is an interesting avenue for future work. However, our observations shouldn’t be over-interpreted, as they may be as much about quirks in Parsel and in the current system.
> >
> > *Table: Additional GPT-3.5 Hypothesis Generation Results*
> > | Method             | 1D-ARC | SyGuS | List Func |
> > |--------------------|--------|-------|-----------|
> > | Direct             | 25.9   | N/A   | 16        |
> > | Program Only (Ours)| 23.1   | 80.9  | 46        |
> > | Full (Ours)        | 26.9   | 86.5  | 57        |
> >
> > > Sec 3.2.2 says summarized hypotheses can often become vague and ambiguous. Will the hypotheses used to guide program generation be of higher quality if you let the model rank the hypotheses? You could analyze the recall@k, i.e., whether top k hypotheses contain a correct one.
> >
> > We have an experiment very similar to this, currently in Appendix B.2 under the paragraph titled “Using LLMs to Rank Hypotheses.” Unfortunately, we were disappointed in the LLM’s ability to rank hypotheses. We specifically followed the approach from “Coder reviewer reranking for code generation” by [2].

---

> ### Author Response · Authors · 2023-11-21
> **Author Response to Reviewer Bu26 (Cont')**
>
> > ARC: In Table 2, using human written hypotheses only has 37.5 accuracy. Does that mean LLM fails to write programs based on correct hypotheses? The statement at the end of page 5 that "GPT-4 is pretty good at both generating hypotheses and realizing them as programs" requires some more evidence or explanation.
>
> First, we agree: we have removed this sentence. For some context, this was intended as a relative statement, given the poor performance of the LLM-based alternatives. We have also added a more explicit we’ve now added a failure case analysis - namely, we were curious about the following: for what subset of the questions did the model fail because it failed to generate a correct hypothesis, and for what subset did it fail because it failed to implement a correct hypothesis? We looked at the 100 sampled tasks and the human-selected correct generated hypotheses (though, we reiterate here that “correct” is both subjective and somewhat arbitrary without experiments in line with those conducted in LARC). For 49% of them, at least one correct hypothesis was identified as correct. Of those selected, 33 (67%) led to correct programs. However, when given all correct language (i.e., the Human-Written Hypothesis evaluation), the model successfully implements 45% correctly. Of these, only one task is solved by the human-selected pipeline that isn’t solved by the human-written pipeline. This may suggest that there is a relationship between the tasks that the model can propose a hypothesis for and the tasks for which it can implement that hypothesis.
>
>
> >  Can you evaluate the recall of model-generated hypotheses, either by some automatic metric or human evaluation?
>
> We have included additional experiments in Appendix B.1 where we evaluate the recall of model-generated hypotheses on multiple datasets. On ARC, we measure the percentage of tasks with correct hypotheses (judged by human) when we increase the number of hypotheses. On 1D-ARC and List Funcctions, we measure the accuracy of our pipeline as we increase the number of hypotheses. Both experiments show that the recall steadily increases as the number of hypotheses increases.
>
> > For ARC, why do you only consider top-1 accuracy but not top-3 as in the official evaluation? Can you compare your method with state-of-the-art methods on the task?
>
> Although top-3 is indeed technically the “official” evaluation metric, top-1 has been the often-unstated convention in LLM-based papers (e.g. [3,4,5]). In general, we would argue that pass@1 makes more sense since we care about evaluating generalization. For some context, we’ve also now highlighted the special-purpose DSL-based state-of-the-art (according to [4]) in our text, which performs marginally worse than our results with human-written hypotheses, solving 43 of our 100 sampled tasks.
>
> > What are the types of tasks that (1) program generation and (2) hypotheses search can be helpful? Can you summarize the features of such tasks? "Inductive reasoning tasks" is too general and abstract. To begin with, is it true that the method is applicable only to symbolic pattern recognition tasks?
>
> In principle, we would argue this framework should be applicable for any inductive reasoning task that can be represented computationally — in practice, there are a few types of inductive reasoning tasks that are not currently supported in our framework.
>
> For example, many more complex visual inductive reasoning tasks will require extending this implementation, such as using a compositional visual programming language to represent the program (e.g., VISPROG [6]). Moreover, if a task is truly unprogrammable, then we lose a core value of this work, namely the inductive bias and generalization that comes with a programmatic representation. Some challenges may also arise for models with a higher false-positive rate which we’ve now noted in the limitations.
>
> In terms of tasks that cannot be solved by Python programs, in discussing future work, we’ve added discussion around more open-ended tasks like summarization where we might want to optimize a “soft” score (e.g., ROUGE [7]). In short, these problems are still approachable with Python but may require the language model to leverage another machine learning model.
>
>
> > Typo: Sec 3.1 "It contains Although simpler..."
> Thank you, this is now fixed!
>
>
> [2] “Coder reviewer reranking for code generation,” Zhang et al. 2023
> [3] “LLMs and the Abstraction and Reasoning Corpus: Successes, Failures, and the Importance of Object-based Representations,” Xu et al. 2023
> [4] “Large Language Models as General Pattern Machines,” Mirchandani et al. 2023
> [5] “Large Language Models Are Not Strong Abstract Reasoners,” Gendron et al. 2023
> [6] “Visual Programming: Compositional visual reasoning without training,” Gupta and Kembhavi 2022
> [7] “ROUGE: A package for automatic evaluation of summaries,” Lin 2004

---

### Official Review · Reviewer_PFzt · 2023-11-05

**Soundness:** 3 good
**Presentation:** 3 good
**Contribution:** 2 fair
**Rating:** 5
**Confidence:** 5

**Summary:**

This paper presents the hypothesis search approach for inductive reasoning. Specifically, hypothesis search first generates multiple hypotheses on the shared transformation rule for the given input-output pairs. Afterward, a subset of hypotheses is selected by humans, or summarized by the LLM. Finally, the LLM generates the Python program given a hypothesis, and the program is executed on the input-output pairs to verify the correctness. They evaluate their approach on ARC, 1D-ARC and SyGuS. Using GPT-4, their approach outperforms the baselines that directly generate the answer or the Python program without hypothesis generation. In particular, they demonstrate that using hypotheses generated by GPT-4 achieves the same performance as using human-written hypotheses.

**Strengths:**

1. Inductive reasoning is an important and challenging problem. This work achieves a notable improvement on ARC and 1D-ARC, showing that combining both abstract hypothesis and concrete code is beneficial.

2. The approach of hypothesis summarization is interesting. Also, it is an interesting finding that using GPT4-generated hypotheses achieves the same performance as using human-written hypotheses, demonstrating the promise of LLMs for generating high-quality hypotheses for inductive reasoning.

**Weaknesses:**

While the overall results are promising, a lot of important ablations and details are missing in the draft.

1. What is the performance with different number of hypotheses? Specifically, in Table 1, it is important to know the performance with fewer number of initial generated hypotheses, such as 8. Comparing hypothesis summarization with directly generating 8 initial hypotheses can validate the importance of the hypothesis summarization stage.

2. In Table 1, the comparison of sample size and token size among different methods is unclear. Specifically, for hypothesis summarization, it is better to uniformly require the model to generate 8 programs for each of the 8 hypotheses for all problems, instead of only applying to 21 tasks, so that the sampling size is more comparable to the program prompting. Similarly, for human-selected hypotheses, it is unclear how many hypotheses are kept after filtering. It is better to always keep 8 hypotheses after filtering. In addition, it is unclear why the number of execution rounds varies for different methods. It is better to unify the setup for a fair comparison.

3. From Table 2, it is interesting to see that the final performance of GPT-3.5 is comparable to GPT-4. Have you tried gpt-3.5-turbo-16k, which has a longer context length? The performance may further improve.

4. The findings on SyGuS are divergent from the main evaluation, as the best result is achieved with purely code generation.

5. Please provide a quantitative analysis on the failure mode; i.e., the percentage of error cases where none of the hypothesis is correct, and the percentage of error cases caused by the wrong generated programs.

6. Please provide the full prompt including the few-shot demonstrations. The appendix only contains the zero-shot prompt. What is the performance of zero-shot prompting? How much does adding 1 or 2 problems in the prompt affects the performance?

7. The evaluation sets of ARC and 1D-ARC are too small. It is better to include at least 100 tasks.

**Questions:**

1. What is the performance with different number of hypotheses?

2. Make the comparison of sample size and token size among different methods clearer. Specifically, for hypothesis summarization, it is better to uniformly require the model to generate 8 programs for each of the 8 hypotheses for all problems, instead of only applying to 21 tasks, so that the sampling size is more comparable to the program prompting. Similarly, for human-selected hypotheses, it is unclear how many hypotheses are kept after filtering. It is better to always keep 8 hypotheses after filtering. In addition, it is unclear why the number of execution rounds varies for different methods. It is better to unify the setup for a fair comparison.

3. For Table 2, have you tried gpt-3.5-turbo-16k, which has a longer context length? The performance may further improve.

4. Please provide a quantitative analysis on the failure mode; i.e., the percentage of error cases where none of the hypothesis is correct, and the percentage of error cases caused by the wrong generated programs.

5. Please provide the full prompt including the few-shot demonstrations. The appendix only contains the zero-shot prompt. What is the performance of zero-shot prompting? How much does adding 1 or 2 problems in the prompt affects the performance?

6. The evaluation sets of ARC and 1D-ARC are too small. It is better to include at least 100 tasks.

---

> ### Author Response · Authors · 2023-11-21
> **Author Response to Reviewer PFzt**
>
> We sincerely thank you for this very in-depth and constructive review!
>
> > What is the performance with different number of hypotheses?
>
> Thanks for this excellent question! We’ve performed additional experiments in Appendix B.1 to investigate the performance as the number of considered hypotheses changes and found that the accuracy of models increases consistently with more hypotheses from GPT-4.
>
> > In Table 1, the comparison of sample size and token size among different methods is unclear. Specifically, for hypothesis summarization, it is better to uniformly require the model to generate 8 programs for each of the 8 hypotheses for all problems, instead of only applying to 21 tasks, so that the sampling size is more comparable to the program prompting.
>
> For hypothesis summarization, we agree that it is better and more rigorous to evaluate on all the tasks instead of a subset. Originally, a subset was used due to cost constraints, but we have now been able to access additional resources. We are currently running this experiment on the full 100 tasks and expect to have results before the end of the discussion period – please note, however, that we do not expect the final results on hypothesis summarization to meaningfully change.
>
> > Similarly, for human-selected hypotheses, it is unclear how many hypotheses are kept after filtering. It is better to always keep 8 hypotheses after filtering.
>
> We will only keep the first correct hypothesis found for each task. So we are testing one hypothesis for each task in the human-selected experiment. We agree that selecting 8 hypotheses after filtering would be better for consistency; there are likely other clever oracles that one might use for filtering for this (e.g., using GPT-4 to find the most similar hypotheses to the ground truth), and we believe this would be a valuable exploration for future work.
>
> > In addition, it is unclear why the number of execution rounds varies for different methods. It is better to unify the setup for a fair comparison.
>
> Great point! We have unified all of the experiments to have three feedback rounds.
>
> > From Table 2, it is interesting to see that the final performance of GPT-3.5 is comparable to GPT-4. Have you tried gpt-3.5-turbo-16k, which has a longer context length? The performance may further improve.
>
> First, we apologize for the confusion - the table is actually showing GPT-4 results, but the caption was ambiguous. We have run an additional experiment using gpt-3.5-turbo-16k to generate 128 programs with human-written hypotheses on the 100 sampled ARC tasks and achieved an accuracy of 30%, which is a little bit higher than the accuracy of gpt-3.5-turbo (27%).
>
> > The findings on SyGuS are divergent from the main evaluation, as the best result is achieved with purely code generation.
>
> Thank you for this point! We’d like to clarify that we don’t consider the SyGuS results as contradicting the overall result: indeed, our pipelines (both Program Only and Full) both perform significantly better than prior work. However, we do find that the natural language hypotheses are not a useful additional abstraction in this dataset. This is likely because the SyGuS problems are relatively easy.
>
> We have also introduced a new dataset to demonstrate the usefulness of natural language hypotheses on non-ARC tasks. Specifically, we now investigate the cognitive-science-inspired inductive reasoning dataset, List Functions, from “The Child as Hacker” [1] which is fairly different from any of the current three datasets. As on the ARC datasets, this shows a clear advantage with the natural language hypotheses.
>
> | Method           | 64 hypotheses   |
> |------------------|-----------------|
> | Direct Prompting | 31%             |
> | Program Only     | 59%             |
> | Full             | 69%             |
>
> [1] “The Child as a Hacker,” Rule et al. 2020

---

> > ### Author Response · Authors · 2023-11-21
> > **Author Response to Reviewer PFzt (Cont')**
> >
> > > Please provide a quantitative analysis on the failure mode; i.e., the percentage of error cases where none of the hypothesis is correct, and the percentage of error cases caused by the wrong generated programs.
> >
> > Great suggestion! For the human-selected evaluation on ARC, we’ve now added a failure case analysis - namely, we were curious about the following: for what subset of the questions did the model fail because it failed to generate a correct hypothesis, and for what subset did it fail because it failed to implement a correct hypothesis? We looked at the 100 sampled tasks and the human-selected correct generated hypotheses (though, we reiterate here that “correct” is both subjective and somewhat arbitrary without experiments in line with those conducted in LARC). For 49% of them, at least one correct hypothesis was identified as correct. Of those selected, 33 (67%) led to correct programs. However, when given all correct language (i.e., the Human-Written Hypothesis evaluation), the model successfully implements 45% correctly. Of these, only one task is solved by the human-selected pipeline that isn’t solved by the human-written pipeline. This may suggest that there is a relationship between the tasks that the model can propose a hypothesis for and the tasks for which it can implement that hypothesis.
> >
> > > Please provide the full prompt including the few-shot demonstrations. The appendix only contains the zero-shot prompt. What is the performance of zero-shot prompting? How much does adding 1 or 2 problems in the prompt affects the performance?
> >
> > Thanks for the suggestion. We have included the full prompt for ARC and 1D-ARC in Figure A.8 and Figure A.9, including the two demonstrations. We test zero-shot prompting on the 1D-ARC dataset, keeping the number of generated hypotheses and programs per hypothesis the same. This gives us an accuracy of 71.3%, which is slightly lower than the performance with 2-shot prompting (73.1%). This shows that examples are helpful but not crucial in hypothesis generation. Note that we do not use demonstrations for the SyGuS and List Functions datasets.
> >
> >
> > > The evaluation sets of ARC and 1D-ARC are too small. It is better to include at least 100 tasks.
> >
> > Thank you for this excellent point! Originally, scaling this up was prohibitively expensive. However, we have now been able to access additional resources, allowing us to extend these results to 100 tasks. We refer the reviewer to the main paper for more details. All the conclusions remain consistent with the previous results, which demonstrate the effectiveness of our proposed pipeline.

---

> ### Author Response · Authors · 2023-11-21
> **Author Response to Reviewer PFzt (Evaluating All Summarized Hypotheses)**
>
> Hi! We appreciate your patience -- we've now also completed the experiment where we evaluated all of the summarized hypotheses, not just the ones where the tasks had human-selected hypotheses. Including the problems without human-selected hypotheses solved two additional problems, bringing the summarized score up to 30% (from 28%). Notably, this suggests that the success rate for problems with a correct hypothesis before summarization was roughly 57% (28/49), while the success rate for the others was about 4% (2/51). We've updated all of the corresponding parts of the paper (Tables 1 and 2, the abstract, and Section 3.2).
>
> For one of the newly solved problems, the summarized hypothesis is correct despite all of its corresponding hypotheses having mistakes. In this case, the summarization removes some incorrect details from the original hypotheses. For the other problem, the model solved it despite not having a correct hypothesis.
>
> Thanks again for the great suggestions, and please let us know if you have any remaining questions!

---

### Official Review · Reviewer_SPJ6 · 2023-11-05

**Soundness:** 2 fair
**Presentation:** 3 good
**Contribution:** 2 fair
**Rating:** 3
**Confidence:** 4

**Summary:**

The paper addresses the challenge of inductive reasoning in large language models (LLMs). Directly prompting by in-context learning may not be able to solve complex tasks. The authors propose a novel approach inspired by the Bayesian rule that involves generating explicit hypotheses in natural language and then translating them into concrete Python programs, which can be verified. This approach, tested on tasks like ARC, 1D-ARC, and SyGuS, significantly improves LLMs' performance. By combining abstract reasoning with programmatic logic, and filtering hypotheses through LLM summaries or human annotators, the method demonstrates substantial improvements, achieving up to 37.5% accuracy on ARC, compared to a 12.5% baseline. The paper highlights the synergy between natural language processing and programmatic approaches in enhancing LLM inductive reasoning.

**Strengths:**

1. The paper introduces a novel method of enhancing inductive reasoning in LLMs by generating explicit hypotheses and translating them into Python programs. This approach creatively combines the strengths of natural language processing and programmatic logic, offering a unique solution to the challenge of inductive reasoning in complex tasks.
2. The paper stands out for its robust methodology and the quality of its experimental results. The authors thoroughly test their approach on challenging datasets like ARC, demonstrating significant improvements in LLM performance. The ablation studies further substantiate the quality of the research, clarifying the contributions of each component of the proposed method.
3. The presentation is good with clarity, presenting complex ideas and methodologies in a comprehensible manner. This clarity enhances the paper's accessibility to a broad audience, which is crucial for disseminating innovative ideas.

**Weaknesses:**

1. While the method of generating and implementing hypotheses as Python programs is innovative, it may pose scalability challenges. For instance, generating a large number of hypotheses for complex problems could be computationally intensive and time-consuming. Moreover, the filtering process—whether automated or human-assisted—might not efficiently narrow down to the most effective hypotheses. To improve, the authors could explore more sophisticated algorithms for hypothesis generation that prioritize efficiency and scalability, possibly through more advanced heuristics or machine learning techniques.
2. The paper demonstrates success in specific datasets like ARC, 1D-ARC, and SyGuS, but it's unclear how well this method generalizes to other types of inductive reasoning tasks, particularly those with differing structures or complexity levels, or even cannot be solved with python programs. Also, the baselines are limited only with direct prompting and ablated baselines, with no baselines from related works. In other words, the range of experimental tasks presented is somewhat limited, potentially restricting the scope of the paper’s conclusions.
3. The hypothesis proposal and selection process is essentially a search problem. The proposed iterative sampling and verification process is costly and inefficient from the perspective of search. The authors could consider more advanced search methods, such as DFS/BFS/MSTC, etc. Get some inspiration from the recent tree search prompting literature, like Tree-of-thoughts, reasoning-via-planning, etc.

**Questions:**

- Is there potential for the proposed method to be generalized across a broader array of tasks beyond those presented in the paper?
- How might this method perform tasks that are inherently difficult or perhaps impossible to encapsulate within a programmable framework?
- Could the authors clarify the missing elements in the appendix that might be pertinent to the paper's methodology or findings?
- Regarding the ARC tasks, what is the average duration, and why do most exceed the 4096 token limit imposed by many LLMs?
- The Direct Prompting baseline, is it just few-shot prompting or Chain-of-thought prompting?

---

> ### Author Response · Authors · 2023-11-21
> **Author Response to Reviewer SPJ6**
>
> Thank you for your detailed and insightful comments. We're happy to have the chance to clear up these points, but please let us know if anything is still unclear.
>
> > While the method of generating and implementing hypotheses as Python programs is innovative, it may pose scalability challenges.
>
> This is an excellent point - indeed, we emphasize the high cost of this approach throughout the paper, as well as some strategies that we took to attempt to mitigate it (e.g., since the primary cost comes from implementing rather than generating hypotheses, our summarization approach substantially reduces the necessary costs). We highlight multiple approaches to improve the scalability by focusing on selecting or extracting a smaller set of hypotheses with the language model; on the other hand, our method can also provide a flexible trade-off between performance and cost by varying the number of hypotheses and programs to search for. This is unachievable by the baseline method that directly prompts the answer.
> Recent work makes the reduced performance from the current search-space-reduction approaches less surprising, emphasizing that even advanced language models struggle to evaluate their own generations without external feedback [1].
> Furthermore, we would like to highlight the flexibility of Python code from a program synthesis perspective. Before LLMs, program synthesis typically required operating over a small DSL with a very restrictive grammar, e.g. regular expressions, string manipulations, or 3D graphics [2,3,4,5]. However, we agree that this is an important point. To emphasize it, we have added a section on limitations and future work, where we discuss this.
> > it's unclear how well this method generalizes to other types of inductive reasoning tasks, particularly those with differing structures or complexity levels, or even cannot be solved with python programs
>
> Thanks for this point! In response, we’ve added a new cognitive-science-inspired inductive reasoning dataset, List Functions, from “The Child as Hacker” [6] which is fairly different from any of the current three datasets. As on the ARC datasets, this shows a clear advantage with the natural language hypotheses.
>
> | Method           | 64 hypotheses   |
> |------------------|-----------------|
> | Direct Prompting | 31%             |
> | Program Only     | 59%             |
> | Full             | 69%             |
>
> We also agree that there are always more datasets that we could include. We’d like to note that datasets like 1D-ARC are designed to vary the complexity and structure of ARC, and SyGus is a very differently structured task. In terms of tasks that cannot be solved by Python programs, in discussing future work, we’ve added discussion around more open-ended tasks like summarization where we might want to optimize a “soft” score (e.g., ROUGE [7]). In short, these problems are still approachable with Python but may require the language model to leverage another machine learning model.
>
> > Also, the baselines are limited only with direct prompting and ablated baselines, with no baselines from related works
>
> Thanks for raising this point. One of the reasons that we were excited to work on this direction is that there is not an extensive body of work that has explored inductive reasoning with language models. We will highlight that certain ablations can indeed be seen as corresponding to applying related works to the inductive reasoning task. For example, the program-only baseline without revisions can be seen as somewhat analogous to the “Program of Thoughts” prompting if applied to the inductive reasoning domain [8]. We’ve also added a language-only baseline that can be seen as analogous to a “Chain of Thoughts” baseline when applied to inductive reasoning [9]. We have also now modified our main text to incorporate this. We’ve also run the state-of-the-art DSL-based approach for ARC on our 100-task subset, finding that it performs slightly below the human-written score, solving 43 of the problems correctly. We’ve added this context in Section 3.2.

---

> > ### Author Response · Authors · 2023-11-21
> > **Author Response to Reviewer SPJ6 (Cont')**
> >
> > > The hypothesis proposal and selection process is essentially a search problem… The authors could consider more advanced search methods, such as DFS/BFS/MSTC, etc. Get some inspiration from the recent tree search prompting literature, like Tree-of-thoughts, reasoning-via-planning, etc.
> >
> > First, one should not consider our approach as competing with alternative search algorithms - indeed, they are complementary. One can actually interpret the iterative revision process as BFS: for each hypothesis implementation node, we expand it, revising each one after all others in the same “depth” have been revised. Unfortunately, other techniques like “Tree-of-thoughts” require the model to be able to assess the quality of a node, which we observed as one of their limitations. It is possible that one could implement an MCTS-inspired approach with access to a trainable language model acting as a reward model, in order to improve its ability to assess quality, but this would be a significantly different (albeit valuable) future work.
> >
> > > Is there potential for the proposed method to be generalized across a broader array of tasks beyond those presented in the paper?... How might this method perform tasks that are inherently difficult or perhaps impossible to encapsulate within a programmable framework?
> >
> > In principle, this framework should be applicable for virtually any inductive reasoning task that can be represented computationally — in practice, many more complex visual inductive reasoning tasks will require extending this implementation, such as using a compositional visual programming language to represent the program (e.g., VISPROG [10]). Moreover, if a task is truly unprogrammable, then we lose a core value of this work, namely the inductive bias and generalization that comes with a programmatic representation. Some challenges may also arise for models with a higher false-positive rate, which we’ve now noted in the limitations.
> >
> > > Could the authors clarify the missing elements in the appendix that might be pertinent to the paper's methodology or findings?
> >
> > Could you please elaborate on what elements you believe are missing? Happy to add any details!
> >
> > > Regarding the ARC tasks, what is the average duration, and why do most exceed the 4096 token limit imposed by many LLMs?
> >
> > We’ve clarified that the primary models we use (in the current experiments, gpt-4-0613) has a context window of 8k tokens. The average size of grids in ARC is (averaged over the 100 tasks we consider) is (10x10). In our experiment, there is only one task in ARC whose prompt exceed the context length of GPT-4, and 14 tasks are longer than the context of GPT-3.5, which is 4,096 tokens.
> >
> > >The Direct Prompting baseline, is it just few-shot prompting or Chain-of-thought prompting?
> >
> > The direct prompting baseline does not require the model to generate intermediate language. We’ve now also added a language-only baseline that can be seen as analogous to a “Chain of Thoughts” baseline when applied to inductive reasoning [5]. We find that, when not generating programs, regardless of whether the model generates the intermediate hypotheses or whether we provide human-written hypotheses, the pass rate is only marginally higher (from 17% without any language to 19% in either language-only case).
> >
> > [1] “Large Language Models Cannot Self-Correct Reasoning Yet,” Huang et al. 2023
> > [2] “RobustFill: Neural Program Learning under Noisy I/O,” Devlin et al. 2017
> > [3] “Write, Execute, Assess: Program Synthesis with a REPL,” Ellis et al. 2019
> > [4] “CSGNet: Neural Shape Parser for Constructive Solid Geometry,” Sharma et al. 2018
> > [5] “From Language to Programs: Bridging Reinforcement Learning and Maximum Marginal Likelihood,” Guu et al. 2017
> > [6] “The Child as a Hacker,” Rule et al. 2020
> > [7] “ROUGE: A package for automatic evaluation of summaries,” Lin 2004
> > [8] “Program of Thoughts Prompting: Disentangling Computation from Reasoning for Numerical Reasoning Tasks,” Chen et al. 2022
> > [9] “Chain-of-thought prompting elicits reasoning in large language models,” Wei et al. 2022.
> > [10] “Visual Programming: Compositional visual reasoning without training,” Gupta and Kembhavi 2022

---

### Official Review · Reviewer_qtss · 2023-11-10

**Soundness:** 3 good
**Presentation:** 3 good
**Contribution:** 2 fair
**Rating:** 6
**Confidence:** 2

**Summary:**

This paper proposes a pipeline to solve abstraction and reasoning tasks. The pipeline prompts LLMs to propose hypothesis about the problem, convert the hypothesis into executable programs, which is later validated against the ground truth outputs given inputs. Experiments on ARC, 1D-ARC, and SyGus demonstrates that the proposed pipeline is effective.

**Strengths:**

- The ablation studies are quite extensive. The authors dissect the effect of each component in the pipeline by, for example, skipping the program generation, skipping generation of natural language hypothesis. The performance improvement of the full pipeline is also clear.
- The abundant technical details contribute to the reproducibility of the work.

**Weaknesses:**

- Which part of the pipeline is novel is not quite clear from the paper writing

The paper introduces every part the proposed pipeline in intensive details - but it is not quite clear which part of the pipeline is novel. I feel compared to earlier works like program-of-thoughts, the novel part is generating natural language hypothesis before program generation and a verification step to verify the correctness of hypothesis. I suggest adding a paragraph in introduction to highlight which parts are novel and the contributions of the work.


- I feel some experiments, such as comparing the performance of GPT 3.5 and GPT 4 is not relevant to the main contribution of the paper. The numbers of these experiments can be moved to appendix to avoid distraction.

**Questions:**

- In Table 3, why are the names of the methods different from Table 1. Does "Full" in Table correspond to any method in Table 1?
- For negative results presented in Sec. 3.4, I suggest to summarize them in a table as well.

---

> ### Author Response · Authors · 2023-11-21
> **Author Response to Reviewer qtss**
>
> Thank you for your excellent questions!
>
> > The paper introduces every part the proposed pipeline in intensive details - but it is not quite clear which part of the pipeline is novel. I feel compared to earlier works like program-of-thoughts, the novel part is generating natural language hypothesis before program generation and a verification step to verify the correctness of hypothesis.
>
> Thank you for this great point! We’ve added a short list of contributions to the introduction.
>
> > I feel some experiments, such as comparing the performance of GPT 3.5 and GPT 4 is not relevant to the main contribution of the paper. The numbers of these experiments can be moved to appendix to avoid distraction.
>
> Thanks again for the suggestion - we’ve now moved the GPT-3.5 experiments to the appendix.
>
> > In Table 3, why are the names of the methods different from Table 1. Does "Full" in Table correspond to any method in Table 1?
>
> This is a great question. Because the problem sizes (in terms of the number of tokens necessary) and the tasks are easier for 1D-ARC, for the “Full” method, we evaluate all of the generated hypotheses. In contrast, a “Full” evaluation of all of the generated hypotheses for ARC would be prohibitively infeasible for us at the current cost of GPT-4. We’ve now clarified this in the text of the corresponding subsection. For the direct prompting method, we report the results from [2] for a direct comparison.
>
> > For negative results presented in Sec. 3.4, I suggest to summarize them in a table as well.
>
> Thank you for this suggestion! We’ve added a table to Sec. 3.4, but we’d like to clarify that we don’t consider the SyGuS results as negative. Indeed, both the Program-Only and Full hypothesis search pipelines perform significantly better than prior work. However, we do find that the natural language hypotheses are not a useful additional abstraction in this dataset. Likely because the SyGuS tasks are simply easier. We also introduce a new dataset to demonstrate the usefulness of natural language hypotheses on non-ARC tasks. Specifically, we now investigate the cognitive-science-inspired inductive reasoning dataset, List Functions, from “The Child as Hacker” [1] which is fairly different from any of the current three datasets. As on the ARC datasets, this shows a clear advantage with the natural language hypotheses.
>
> | Method           | 64 hypotheses   |
> |------------------|-----------------|
> | Direct Prompting | 31%             |
> | Program Only     | 59%             |
> | Full             | 69%             |
>
>
> [1] “The Child as a Hacker,” Rule et al. 2020
> [2] LLMs and the Abstraction and Reasoning Corpus: Successes, Failures, and the Importance of Object-based Representations. Xu et al. 2023

---

### Author Response · Authors · 2023-11-21
**General Response**

We thank all reviewers for their thoughtful comments and helpful suggestions. We appreciate that reviewers felt the work was “novel” (**SPJ6**) and “original” (**e7wF**). Moreover, we appreciate that they identified the ablations as “extensive” (**qtss**) and felt that the work was reproducible due to its “abundant technical details” (**qtss**) and “robust methodology” (**SPJ6**). We also appreciate that it was highlighted that this work is “important to the program synthesizing community” (**e7wF**), as well as the recognition of the challenging datasets we focus on (**7fhN**). Further, we are glad that so many reviewers had positive comments on the presentation, “presenting complex ideas and methodologies in a comprehensible manner” (**SPJ6**) and clear (**Bu26**). Lastly, we thank the reviewers for describing the improvements as “notable” (**PFzt**), “significant” (**SPJ6**) and “promising” (**e7wF**).

We also address the constructive experimental points made by the reviewers with new experiments, including points about the experiment size, the breadth of datasets, and the role of the number of hypotheses. In response to comments about the clarity of the contributions, limitations, and failure analysis, we have also added the corresponding writing.

We have made revisions to our paper accordingly, which are highlighted in blue in the updated submission. The main changes include

Experiments:
1. Scaling up experiments (Section 3.2, 3.3): multiple reviewers (**PFzt**, **7fhN**) suggested that the number of tasks evaluated on ARC was too small – as a result of additional resources becoming available to us, we have been able to extend our experiments on ARC and 1D-ARC to 100 and 108 tasks. Our conclusions remain consistent with the scaled-up results, demonstrating the effectiveness of our method.
2. A new dataset (Section 3.5): reviewers asked about generalization to more non-ARC datasets (**SPJ6**, **7fhN**), so we added the List Functions list transformation dataset, on which our method again demonstrates significant improvement.
Ablation on number of hypotheses (Appendix B.1): reviewers noted the value in understanding the impact of different numbers of hypotheses across our datasets, so we have conducted these new evaluations (**PFzt**, **e7wF**, **SPJ6**), showing that performance consistently improves when we increases the numer of hypotheses sampled.
3. Experiments with GPT-3.5 (Appendix B.2): there were multiple questions raised about the performance of GPT-3.5, especially in the context of its poor hypothesis generation ability on ARC (**Bu26**, **7fhN**). We ran the full hypothesis search pipeline on GPT-3.5, finding that on 1D-ARC, List Functions, and SyGus, the results are consistent with the GPT-4 results. We also added experiments with GPT-3.5-16k (**PFzt**), finding a small improvement on ARC with human-written hypotheses (27% -> 30%). Lastly, as suggested by **qtss**, we have moved these results to the appendix B.2.

Writing:
1. Contributions list: multiple reviewers (**qtss**, **SPJ6**, **7fhN**) indicated that a clear list of contributions would be helpful - we have added one in the introduction. We hope the addition of this list will help clear up any misunderstandings on the core contribution of our work, which lies in inductive reasoning with language models by generating and testing hypotheses at multiple levels of abstraction, as opposed to, e.g., in self-revision (**7fhN**).
2. Limitations and future work: some reviewers (**e7wF**, **SPJ6**, **Bu26**) highlighted limitations or potential future directions - we have added these to a new section on limitations and future work.
3. Quantitative failure mode analysis: reviewers (**e7wF**, **PFzt**, **Bu26**) asked for an analysis of when the failures on the human-selected attempts were due to incorrect hypotheses and when they were due to a failure by the language model to implement the hypotheses. We’ve added this.

We have revised the PDF accordingly and responded to each point raised individually in the comments below. Feel free to let us know if you have any additional questions!

---

### Meta-Review · Area_Chair_F3Fm · 2023-12-05

**Metareview:**

The paper studies inductive reasoning with language models. Specifically, they prompt LLM to propose abstract hypotheses, map hypotheses into executable programs, which is later validated against the ground truth outputs given inputs by LLM or human. The approach is evaluated on multiple benchmarks (ARC, 1D-ARC, and SyGuS) on two domains (code generation and synthetic reasoning dataset), and shown to improve LLMs' performance substantially on both. The paper is overall well written. One major weakness of the approach is that it can be inefficient as it requires generating multiple hypotheses.

**Justification For Why Not Higher Score:**

The efficiency of the proposed method should be evaluated more clearly. The method performs better, but these comes with more compute, and those should be *quantitatively* evaluated. I strongly encourage authors to add this section.

**Justification For Why Not Lower Score:**

The paper explores an interesting idea, and experiments are solid.

---

### Decision · Program_Chairs · 2024-01-16

Accept (poster)